# Common and distinct neural correlates of social interaction processing and theory of mind in narratives

Zizhuang Miao [1], Heejung Jung [1], Philip A. Kragel [2], Ke Bo [1], Patrick Sadil [3], Martin A. Lindquist [3] & Tor D. Wager [1] ✉

Social interaction processing and theory of mind (ToM) frequently co-occur, but their commonalities and distinctions at behavioral and neural levels remain unclear. Participants ($N = 231$) provided moment-by-moment ratings of four text and four audio narratives on social interactions and ToM engagement, which were reliable (split-half $r = 0.98$ and 0.92, respectively) but only modestly correlated ($r = 0.32$). In a second sample ($N = 90$), we analyzed the covariation between social interaction and ToM ratings and fMRI activity during text and audio narratives. Activity maps associated with social interaction processing and ToM generalized across text and audio (spatial $r = 0.60$ and 0.58, respectively) and overlapped in canonical ToM regions (FDR $q < 0.01$). ToM uniquely engaged the anterior intraparietal sulcus, right lateral occipitotemporal cortex, and right supplementary motor area. These results suggest that observing social interactions automatically engages canonical ToM regions, even without explicit mentalizing, and ToM additionally engages brain regions related to action understanding.

As social animals, we are surrounded by others whose appearances, expressions, and actions we can perceive and whose thoughts, beliefs, and emotions we must infer to successfully navigate the world. Based on dual-process accounts of social cognition[1–3], there are two broad types of social cognitive processes: (1) perceptual processes that are automatic, inflexible, and specific to a sensory modality, and (2) conceptual processes that are controlled, flexible, and operate on multimodal information. This framework has motivated a separation in social neuroscience research, with some studies focusing on automatic social perception[4] and others on controlled social inferences[5]. Thus, separate literature addresses social interaction processing–the recognition or the emergence of the perception of other agents' interactions from a third-person perspective[6–10]–and theory of mind (ToM)–the general mental process of inferring and understanding other agents' mental states[11–14]. This separation has obscured our understanding of (1) the neural systems involved in conceptual aspects of social interaction processing, (2) the degree to which social

interaction processing and ToM covary in naturalistic settings, and (3) their shared and unique neural correlates.

First, analogous to other types of perception, social interaction processing involves both perceptual and conceptual processes. As with recognizing an object in a two-tone Mooney image[15], which is facilitated by prior knowledge of the object's identity[16,17], perceiving social interactions requires matching conceptual representations stored in memory with higher-order statistics computed from the input[18,19], such as the movement patterns of geometric shapes[10,20], point-light displays[8], and human figures[21,22]. However, nearly all existing studies have used visual or audiovisual stimuli to elicit recognition of social interactions (for other examples see refs. 23–28). This makes it unclear whether regions like the superior temporal sulcus (STS) are involved in sensory modality-specific perceptual or transmodal conceptual aspects of social interaction processing, or both. STS is sensitive to both visually and auditorily presented social interactions[6] and has shared neural representations of social interactions across movie

[1]Dartmouth College, Hanover, NH, USA. [2]Emory University, Atlanta, GA, USA. [3]Johns Hopkins University, Baltimore, MD, USA.
✉e-mail: Tor.D.Wager@dartmouth.edu

watching and free recall[29], suggesting its important role in hetero-modal integration of social interaction information and potentially in transmodal conceptual aspects of social interaction processing. To isolate the conceptual processes, verbal narratives can be an effective approach because they lack bottom-up sensory input over which higher-order statistics related to social interactions can be directly computed. Recognizing social interactions in narratives must be mediated by semantic concepts. Given related findings that verbal descriptions of objects[30] or actions[31,32] activate overlapping brain regions with visual presentations of the same objects or actions, we hypothesized that linguistic narratives describing social interactions would engage some brain regions shared with visually and/or auditorily presented social interactions, particularly in STS.

Turning to inferences about mental states, ToM has traditionally been construed as a controlled and deliberative process[11,12,33] and studied by neuroimaging tasks that require explicit reasoning about false beliefs[13] and causal inferences about others' actions[14]. However, ToM may also include automatic or even unconscious computations[34,35], and brain regions responding to controlled ToM tasks can be engaged by features of movies defined as ToM events[36–38]. These results suggest that canonical ToM regions can be activated by passively experiencing stimuli with certain features. Social interactions could be a key feature that naturally invites the use of ToM[39–41], but few studies have provided evidence for this. Here, we tested to what extent social interactions covary with the use of ToM in the same naturalistic narratives and whether they activate regions involved in ToM.

In some prior studies, social interaction processing and ToM activate separate regions with no overlap[8,9]. However, those findings may confound processes with stimulus types because they do not match social interaction content when assessing ToM, and vice versa. For example, the false belief stories typically used to study ToM[13] have more social interactions than control stories, raising the possibility that some activity in ToM-associated regions is attributable to social interaction processing. Conversely, animations depicting social interactions[10,20] also induce ToM[42], raising the possibility that some social perception-related activity is attributable to ToM. Indeed, other evidence suggests overlapping neural correlates of social interaction processing and ToM, including in posterior STS (from a cross-stimulus meta-analysis, see ref. 43) and dorsomedial prefrontal cortex (dmPFC[5,26]). Our use of narratives to induce both social interaction processing and ToM can better rule out potential confounds. If social interaction processing affords automatic engagement of ToM, we would expect to observe common neural correlates even in the absence of explicit demand on or experience of ToM. In addition, ToM may be more demanding and engage these common processes to a greater degree. Finally, if automatic recognition of social interactions and deliberative ToM are neurally separable, as some studies suggest, both may involve unique activations.

Although a couple of recent studies have directly compared the neural correlates of social interaction processing and ToM in the same task[44,45], there is a methodological limitation worth noting: they operationalize social interactions and ToM in movies solely by stimulus features. For example, ToM has been defined as moments in movies in which the characters are inferring other characters' mental states[44,45], or as moments in which viewers are led to think about the characters' thoughts[36–38]. These definitions capture affordances in stimuli for mental state inferences and the need to enage ToM, which can be termed ToM demands. However, defining social interactions and ToM in terms of researcher-coded stimulus features may not match people's actual moment-by-moment processing or their underlying cognitive processes[46], reducing validity and sensitivity in identifying their neural correlates[10]. To overcome this limitation, we correlated fMRI activity with self-report measures of moment-by-moment recognition of social interactions and use of ToM during narratives.

Taken together, we aimed to use verbal narratives to selectively engage the conceptual aspects of social interaction processing via verbal narratives and to investigate whether it activated shared brain regions with ToM. By linking the two processes to neural activity elicited by the same stimuli, we sought a better understanding of their individual neural correlates, as well as the commonalities and differences between them. We took a cross-modal replication strategy by presenting narratives in both text and audio forms and identifying common brain regions that encoded social interactions or ToM across the two sensory modalities. We expected this to produce more generalizable and conservative neural results regardless of lower-level stimulus features.

## Results

We collected and analyzed two datasets in the current study. The first dataset came from an online behavioral experiment in which participants ($N = 231$) provided moment-by-moment ratings of the extent to which they thought there were social interactions ($N = 114$) or were themselves engaging in ToM ($N = 117$) as they read four written narratives and listened to another four. We instructed participants to define social interactions as parts of stories that have more than one character and in which one character communicates with or directs actions toward other characters. We instructed them to define ToM as the time points when they are thinking about characters' mental states. We provided instructions and examples to participants before the experiment based on these definitions. We then extracted the medians of participants' ratings to serve as continuous estimates of the time courses of subjective processing of social interactions and engagement of ToM for the general population. The second dataset came from a neuroimaging experiment in which another group of participants ($N = 90$) passively experienced the same narratives during functional magnetic resonance imaging (fMRI) scanning without providing any ratings. We correlated the medians of online behavioral ratings with fMRI data using voxel-wise random-effect general linear models (GLMs; Fig. 1a). All inferential statistical tests were two-tailed. Multiple comparisons in voxel-wise modeling of neural data were corrected using false discovery rate (FDR)[47] correction with $q < 0.01$. Other multiple comparisons were corrected using Bonferroni correction.

In the following sections, we first validate our methods by testing whether written narratives and self-report measures reliably reveal the neural correlates of social interaction processing and ToM. We then directly compare the neural correlates of social interaction processing and ToM and use Bayes Factors (BF) to classify each voxel as responding to both social interactions and ToM or only to one. We found that the neural correlates of social interaction processing in narratives overlapped with those found in visual and auditory stimuli. Furthermore, the voxels that responded to social interactions across modalities were also associated with ToM, whereas ToM was uniquely associated with neural activity in other voxels.

### Processing social interaction in narratives activated overlapping regions with audiovisual stimuli

In the online study, participants' ratings were similar across individuals (Fig. 2a, left), as indicated by moderate to high pairwise correlations (median = 0.47, median absolute deviation [MAD] = 0.11). Participants most frequently used the two ends of the scale to answer the social interaction question (Fig. 2a, right), suggesting high confidence about the presence or absence of social interactions. More importantly, the group median time courses were highly reliable, as suggested by large split-half Pearson correlations ([0.96, 0.99], mean ($M$) = 0.98, standard deviations (SD) = 0.002; estimated from 2000 permutation samples and corrected using the Spearman−Brown formula[48,49]. Overall, these results suggest that participants consistently and reliably recognized and reported social interactions.

**a** Experiments and analysis overview

**b** Narrative excerpt with ratings/annotations

**Fig. 1 | Study overview. a** The stimuli used in this study were eight narratives, with the first four presented as audios and the last four as texts. One group of participants ($N = 90$) experienced the narratives during fMRI scanning. Another group of participants ($N = 231$) experienced the narratives online on their personal computers and simultaneously provided moment-by-moment ratings about social interactions in the narratives ($N = 114$) or their use of ToM ($N = 117$). The black rectangles illustrate what participants saw when narratives were presented. Only one of the two questions shown in this figure was presented to any given online participant throughout the study. We regressed blood-oxygen-level-dependent (BOLD) signals from fMRI scanning (a simulated BOLD time series was plotted for illustration only) on the median time series from online ratings (actual data shown here) using voxel-wise GLMs. Shaded areas around the lines indicate medians $\pm 1.96 \times$ (median absolute deviations)/(square root of valid ratings at each time point). **b** An example narrative excerpt extracted from the eighth trial of Narrative #1 (see Supplementary Materials for all narrative texts). The medians of social interaction and ToM ratings during this excerpt, as well as researchers' annotations of ToM demands (see Validation of self-report measurement in Results for details), were displayed as the highlight color of each word. The approximate time window of this excerpt was shown by the horizontal gray bar in (**a**).

Next, we tested whether social interactions in narratives activated regions similar to those found in audiovisual stimuli. We used the medians of online participants' ratings in the Text and Audio modalities separately to regress each fMRI participant's neural activity. We then performed group-level one-sample $t$-tests ($N = 90$) over the individual regression coefficients for the two modalities separately. The effect maps were similar across modalities (Pearson's correlation between two unthresholded whole-brain maps, or spatial $r$, = 0.60; for thresholded maps, see Fig. 3a, left). In a side analysis, we found that whole-brain predictive models for social interactions generalized across modalities (Fig. S2). This suggests that participants use overlapping neural circuitry to process social interactions both when reading and when listening to the narratives, consistent with the modality-independent nature of conceptual processing.

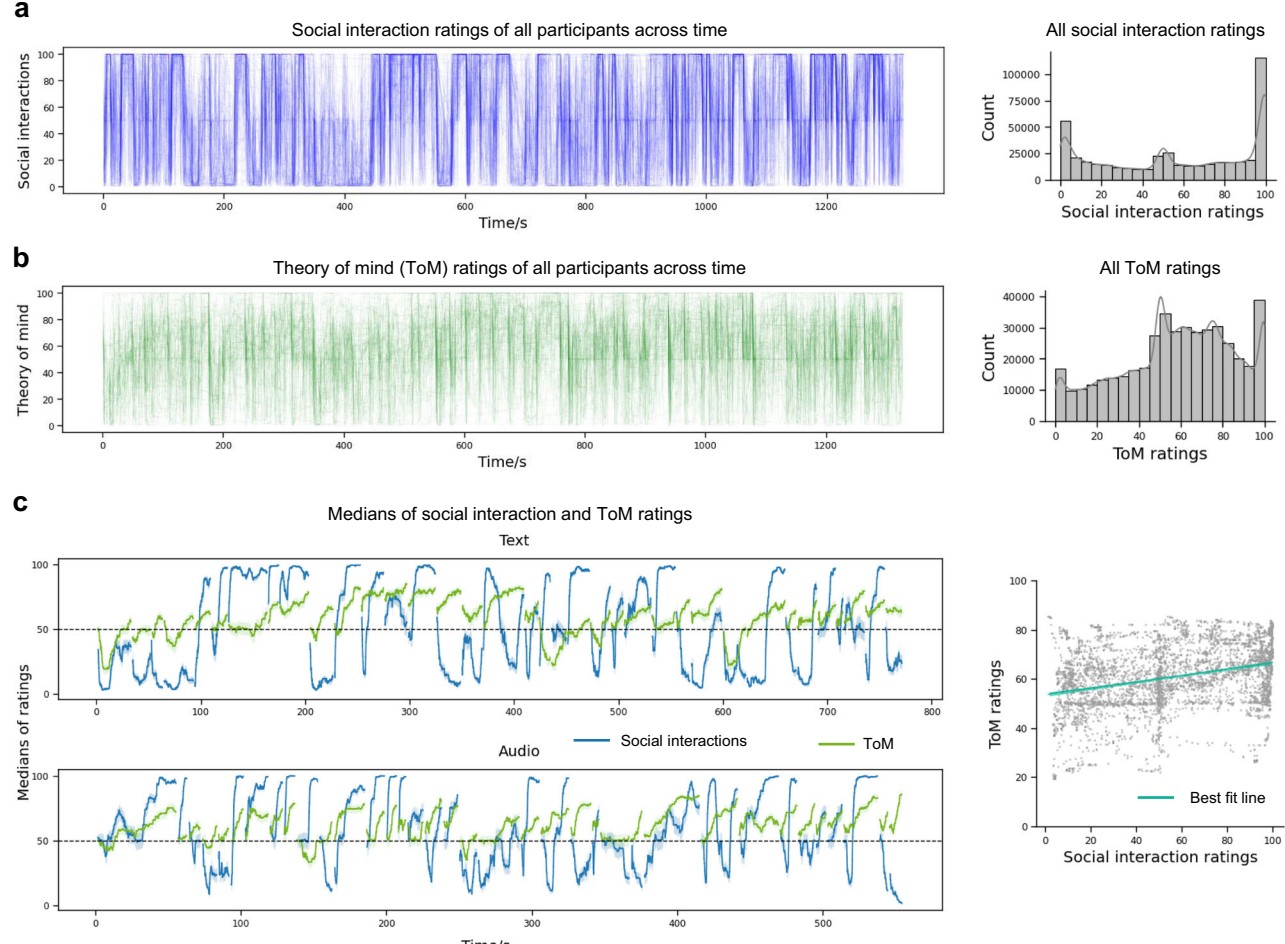

**Fig. 2 | Ratings of online participants about social interactions and ToM.**
**a, b** Left: moment-by-moment ratings of individual participants are plotted as thin, quasi-transparent lines; thicker/darker segments indicate greater overlap across participants at those time points. Right: histograms show the distributions of all ratings. **c** Left: median ratings are shown as solid lines, with shaded error intervals as in Fig. 1. Ratings are split by modality into two plots. Note that lines show discontinuities because a few data points at the start of each trial were removed due to insufficient numbers of participants with valid ratings (see the "Methods" section). Right: the scatter plot shows each time point in the line plots as a dot; the cyan line is the best-fit line estimated by ordinary least squares regression ($r = 0.32$). This figure shows that participants' ratings of social interactions and ToM were consistent, reliable, and only moderately correlated.

We then located the cross-modal neural correlates of social interaction processing by taking the conjunction of the two effect maps for text and audio presentation (Fig. 3a, right). In the conjunction map, a voxel was significant if and only if it was significant in both original maps with the same sign. Values indicated by colors in Fig. 3a are averages of *t*-values for text and audio. Many regions implicated in social cognition studies were significant in the conjunction map, including bilateral superior temporal sulcus (STS) that extends posteriorly into angular gyrus and supramarginal gyrus (temporoparietal junction (TPJ), following the naming convention in[13]), precuneus (PC), portions of medial superior frontal gyrus (dmPFC, following the naming convention in[26]), and inferior frontal gyrus (IFG). Moreover, early visual cortex, including V3 and V4, lateral occipital cortex (LOC), and portions of inferior temporal cortex (ITC)[50] were also significant (Fig. 3a, right; see also Table S6). This suggests that social interactions in narratives are correlated with neural activity in regions typically seen as responsible for visual processing, even when there is no visual information about social interactions. To facilitate comparisons between the current results and known functional divisions of the brain, we correlated effect maps with each of seven canonical functional networks obtained from resting-state fMRI data[51]. The conjunction map of social interaction processing overlapped mostly with the Default Mode Network

(point-biserial $r = .37$) and the Frontoparietal Network (point-biserial $r = .08$).

The cross-modal regions responding to social interactions in the narratives overlapped with those reported in previous studies[26–28]. To further compare the results with previous findings quantitatively, we used the Neurosynth database[52] to identify brain masks where all voxels were significantly related to the appearances of "social interaction", "social cognition", or "psts" (short for posterior STS[4,8,45]). We then extracted the mean regression coefficients (betas) for social interactions within the masks, a metric reflecting the association between within-mask neural activity and behavioral ratings of social interactions. The brain-behavior associations under both modalities within all three masks were significantly above zero (all $Ms > 0.027$, SDs = [0.047, 0.052], all $t(89)s > 5.5$, all $ps < 0.001$, all Cohen's $ds > 0.58$, all lower bounds of the 95% confidence intervals [CI] > 0.017; Fig. 3b). This indicates that brain areas associated with social interactions in previous studies are also activated by the social interaction ratings in the current study.

We also investigated the neural correlates of manual experimenter annotations of social interactions following the methods of previous studies[45,53]. Annotations of social interactions were similar to continuous ratings of social interactions (Pearson's $r = 0.58$) and had similar neural correlates (see Supplementary Results). Annotations

**a**  Group-level effect maps of online ratings of social interactions

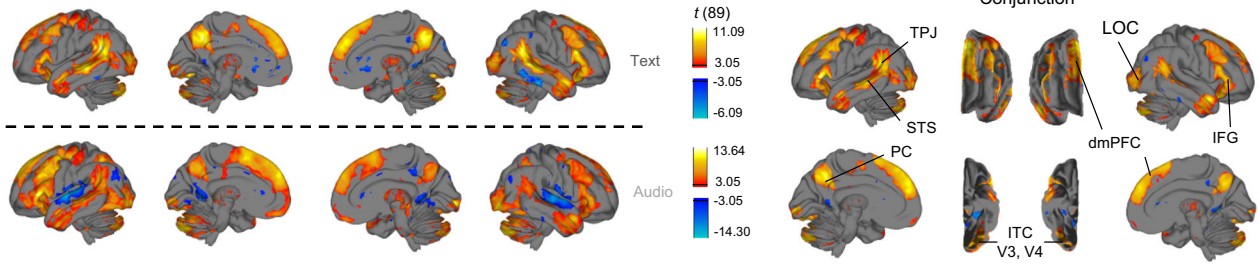

**b**  Average brain-behavior associations within selected brain masks

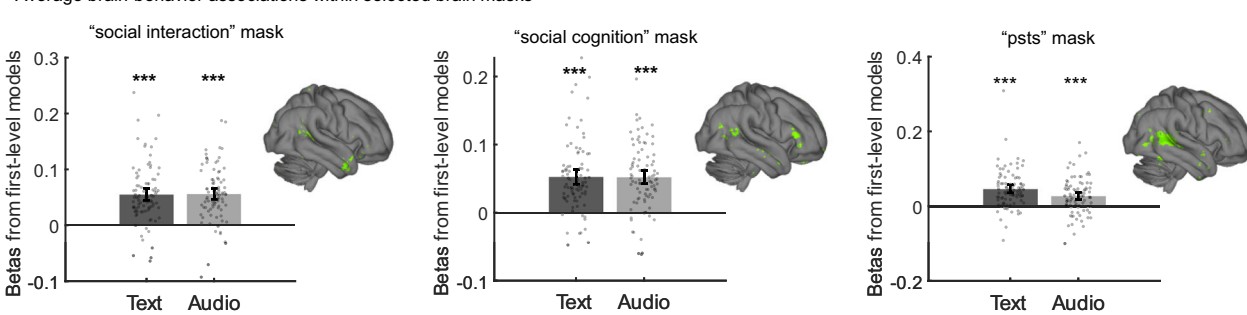

**Fig. 3 | Neural correlates of social interaction processing in narratives. a** Group-level effect map of social interactions as the medians of online participants' ratings, thresholded at FDR[47] $q < 0.01$. Maps on the left show separate patterns for the Text and Audio modalities, and maps on the right show the conjunction across modalities. Analyses were conducted in volumetric space, and results were projected for visualization onto the pial surface used in ref. 50 using the MNIsurf method[93], and onto the thalamus surface[94], brainstem, and cerebellum surface[95] using linear interpolation. Map colors reflect the $t$-values estimated from one-sample $t$-tests against zero. Several key regions that were significant in both hemispheres and both modalities are annotated. TPJ temporoparietal junction, STS superior temporal sulcus, PC precuneus, dmPFC dorsomedial prefrontal cortex, IFG inferior frontal gyrus, LOC lateral occipital cortex, ITC inferior temporal cortex. **b** Group-level effects of social interactions in both modalities within three brain masks from

Neurosynth ($N = 90$ participants for each bar). Regions from each mask are illustrated in green in the right-hemisphere surface plots; the full masks were used for analysis. Bar heights indicate mean regression coefficients (betas) from the first-level models within each mask across participants, which reflect the strength of the associations between social interaction ratings and neural activity within each mask. Error bars indicate 95% confidence intervals, centered around group means. Points show the mean betas for each participant. Each $p$ value was obtained from a two-sided one-sample $t$-test against zero, and all $p$ values were below 1e-13, except the right-most two $p$ values were 5.50e-13 and 2.38e-7. ***$p < 0.001$ after Bonferroni correction of six comparisons. This figure shows that ratings of social interactions were significantly associated with neural activity in widespread brain regions, including visual cortices and canonical social cognition regions, as well as in predefined brain masks associated with social interactions.

---

also served to break down the social interaction content into approximately equal amounts of verbal interactions and physical, non-verbal interactions[21,54], suggesting relatively balanced interaction types (see Supplementary Results). In summary, the results at both behavioral and neural levels indicate that narratives are valid and sensitive stimuli for probing the neural correlates of the conceptual aspect of social interaction processing.

**Validation of self-report measurement of ToM**

In the Introduction, we argued that researchers' annotations of ToM should be treated as stimulus properties and may not be sufficiently sensitive to localize ToM-related neural activity. Those ToM annotations corresponded to stimulus sections with affordances or needs for people to use their ToM[36–38,44,45], and we defined such sections as having ToM demands and annotated them as where in the narratives there are explicit mentions of characters' mental states. To assess how closely ToM demands track the actual use of ToM, we also asked human participants to report how much they were thinking about characters' mental states, or using ToM, while they were experiencing the narratives. We refer to this self-report measurement as ToM engagement. In the current report, mental states include both the cognitive aspects (e.g., beliefs and goals) and the affective aspects (e.g., emotions and desires). Below, we quantitatively compare the two measures and their neural correlates.

At the behavioral level, participants used the middle to right portion of the scale to rate ToM engagement more than social

interactions, as suggested by denser distributions in the 50-80 range (Fig. 2b) and smaller variance across time (mean SD across participants = 19.35 for ToM engagement; 32.97 for social interactions). Participants were less similar to one another on their ToM engagement ratings, as seen in lower pairwise Pearson's correlations between participants (median = 0.21, MAD = 0.15) than in social interactions. Still, the ToM engagement ratings were highly reliable (split-half Pearson's correlations [0.86, 0.95] across 2000 permutation samples ($M = 0.92$, SD = 0.01), corrected using the Spearman–Brown formula[48,49]), supporting the validity of applying those ratings to another participant group.

To obtain ratings for ToM demands, four researchers in our group independently annotated the narratives. We first cut the narratives into sentence parts, the smallest chunk in a sentence that contains one predicate or a sequence of predicates and their associated objects[53] (see Supplementary Materials for the full list of sentence parts). The raters then read the sentence parts in their original order and determined whether each sentence part explicitly mentioned characters' mental states (1 = yes, 0 = no), providing labels for ToM demands (see Fig. 1b for an example). Overall, the raters provided moderately similar annotations about ToM demands (pairwise Pearson's correlations [0.51, 0.60], $M = .57$; Fleiss' kappa = 0.54). We then grouped all the annotations and assigned a consensus label to each sentence part (conflicts were addressed by discussion).

We compared the ToM demand labels to the median time course of ToM engagement ratings in the Text and Audio modalities separately. Correlations between the two measures were not significant

(Text modality: Pearson's $r = 0.14$, $p = 0.22$, 95% CI of the null distribution = [−0.17, 0.18]; Audio modality: $r = 0.19$, $p = 0.09$, 95% CI of the null distribution = [−0.19, 0.19]; $p$ values and 95% CIs were estimated from 10,000 bootstrapped phase-randomization samples). This indicates that annotations of ToM demands capture different behavioral profiles than participants' usage of ToM.

We then investigated whether ToM engagement ratings or ToM demand labels are better at locating neural activity related to ToM, by comparing the neural correlates of the two measures both qualitatively (comparing which regions were significant) and quantitatively (within brain masks associated with ToM in previous studies). Specifically, we fit two GLMs with the only difference being whether the ToM regressors were ToM engagement or ToM demands, and compared neural results across sensory modalities. The neural correlates of ToM engagement were moderately similar across text and audio modalities (spatial $r = 0.58$), as were those of ToM demands (spatial $r = 0.52$). For the qualitative comparison, we found that ToM engagement was significantly correlated with neural activity in bilateral TPJ, STS, PC, dmPFC, and IFG under both modalities, the typical regions associated with ToM[5]. The cross-modality conjunction map of ToM engagement overlapped mostly with the Default Mode Network (point-biserial $r = 0.27$) and the Frontoparietal Network[51] (point-biserial $r = 0.17$). By contrast, ToM demands showed cross-modality activations in only a subset of those regions, including bilateral PC and STS, left TPJ, and left IFG, and the areas of significant regions were smaller when using the same statistical threshold (Fig. 4a). Additionally, the activation patterns of ToM demands were less similar to the Default Mode Network (point-biserial $r = 0.12$) or Frontoparietal Network (point-biserial $r = −0.03$), while more similar to the Visual Network (point-biserial $r = 0.08$), suggesting functional dissociations with ToM engagement.

For quantitative comparisons, we selected brain masks from the "tom" and "mentalizing" term maps in the Neurosynth database[52] and the ToM group maps from a false belief task[55]. The first quantitative comparison was between the brain-behavior associations within masks, operationalized as the group-level effect sizes, Cohen's $d$ (each participant's mean regression coefficients (betas) within masks divided by the standard deviation across participants). To test whether the differences in effect sizes were significant, we used bootstrap tests with 10,000 samples to estimate $p$ values for the null hypothesis that the differences were zero. In both "tom" and "mentalizing" masks, ToM engagement had larger effect sizes than ToM demands in the Audio modality and when averaged across modalities (all effect size differences > 0.84, all $p$s < 0.001, all lower bounds of the 95% CIs of effect size differences > 0.31); there was no difference in the Text modality (both effect size differences < 0.24, both $p$s > 0.05, both 95% CIs included zero). Similar results were observed in the DMPFC and RSTS masks, whereas in the TPJ and PC masks, the differences were significant in both modalities and on average (results from the Neurosynth masks are shown in Fig. 4b; other results are in Fig. S4). This indicates that ToM engagement is associated with neural activity in most canonical ToM regions more strongly than ToM demands.

The second quantitative comparison was on the similarity between the spatial patterns of betas and meta-analytic maps within masks. In each mask created from the Neurosynth database, we extracted each participant's betas and correlated them with the $z$ values in the meta-analytic map. We then transformed Pearson's correlation coefficients to Fisher's $z$ values to ensure normality, which were then subjected to paired-sample $t$-tests. Betas from ToM engagement had a larger spatial pattern correlation with meta-analytic maps (all mean differences > 0.048, all $t(89)$s > 3.9, all $p$s < 0.001, all Cohen's $d$ > 0.42, all lower bounds of the 95% CIs of mean differences > .024; Fig. 4c), except in the "mentalizing" mask in the Text modality (mean difference = 0.009, $t(89) = 0.73$, $p = 0.47$, Cohen's $d = 0.077$, 95% CI of the mean difference = [−0.015, 0.033]). This analysis could not be performed on the ToM group maps[55] because $t$-

values were not available for all voxels. These results indicate that the neural activity associated with ToM engagement is more similar to existing results on ToM. Together, there is strong evidence that ToM engagement has greater statistical power for localizing ToM-related brain regions than ToM demands.

## Comparison between social interaction processing and ToM

So far, we have shown that group-level summaries (medians) of online participants' ratings can serve as a reliable and sensitive measure for identifying the neural correlates of social interaction processing and ToM in the narratives. The medians of the two ratings showed low to moderate correlations (Pearson's $r = 0.29$ for Text modality, $r = 0.37$ for Audio modality, $r = 0.32$ for all time points; Fig. 2c). This supports the view that people tend to use ToM while perceiving social interactions during natural experience, whereas the two processes can also be disentangled. Most sentences have higher or lower ratings for both social interactions and ToM, such as the one below:

> When Linda arrived, she denied having made a move on Dan.

Examples of narrative sentences that have higher social interaction ratings and lower ToM ratings, and vice versa, are shown in Fig. 1b.

We fit a GLM with both the medians of self-reported social interactions and ToM, with separate regressors for the Text and Audio modalities. The correlations between social interaction and ToM ratings remained similar after convolution with the canonical hemodynamic response function (Pearson's $r = 0.29$ for Text modality; $r = 0.38$ for Audio modality), which did not raise concerns about multicollinearity (variance inflation factors < 1.3 for all regressors in the model).

Between the Text and Audio modalities, the effect maps of social interaction processing, when controlling for ToM, were less similar than when modeled alone (spatial $r = 0.49$, Fig. 5a, left); similarly for the effect maps of ToM (spatial $r = 0.51$, Fig. 5b, left). Still, we took the conjunction across modalities (Fig. 5a, b, right) because common neural correlates across modalities are more invariant to low-level stimulus features and thus more specific to the mental processes being compared. The brain regions responding to social interactions under both modalities include bilateral PC, STS, IFG, dmPFC, and left TPJ; those responding to ToM include bilateral TPJ, PC, IFG, dmPFC, and left STS, as well as bilateral anterior intraparietal sulcus (aIPS), supplementary motor area (SMA), and bilateral lateral occipitotemporal cortex (LOTC), which encompasses the middle temporal and medial superior temporal area (MT+). The whole-brain conjunction map of social interactions overlapped predominantly with the Default Mode Network (point-biserial $r = 0.39$), while that of ToM overlapped with both the Default Mode Network (point-biserial $r = 0.22$) and the Frontoparietal Network[51] (point-biserial $r = 0.18$). We also tested the effect sizes of social interactions and ToM against zero in one-sample $t$-tests in selected cortical regions[50] (Table 2), namely left TPJ, bilateral dmPFC, bilateral aIPS, right MT+ (Fig. 5c), bilateral PC, left STS, right TPJ, and bilateral SMA (Fig. S5). In some regions, such as left TPJ and bilateral dmPFC, both social interactions and ToM had significant effects under both modalities (all mean effect sizes > 0.41, all $t(89)$s > 3.8, all $p$s < 0.001, all lower bounds of the 95% CIs > 0.011), whereas in some other regions, such as bilateral aIPS and right MT + , only ToM had significant effects (all mean effect sizes > 0.46, all $t(89)$s > 4.4, all $p$s < 0.001, all lower bounds of the 95% CIs > 0.030). These results imply that social interactions and ToM have largely overlapping neural correlates, and both independently contribute to the neural activity in those regions. At the same time, ToM appears to recruit brain regions that social interactions do not.

To test the robustness of these results to other stimulus features, we quantified a series of features and tested whether any could be alternative explanations for, or confounders of, the neural effects of

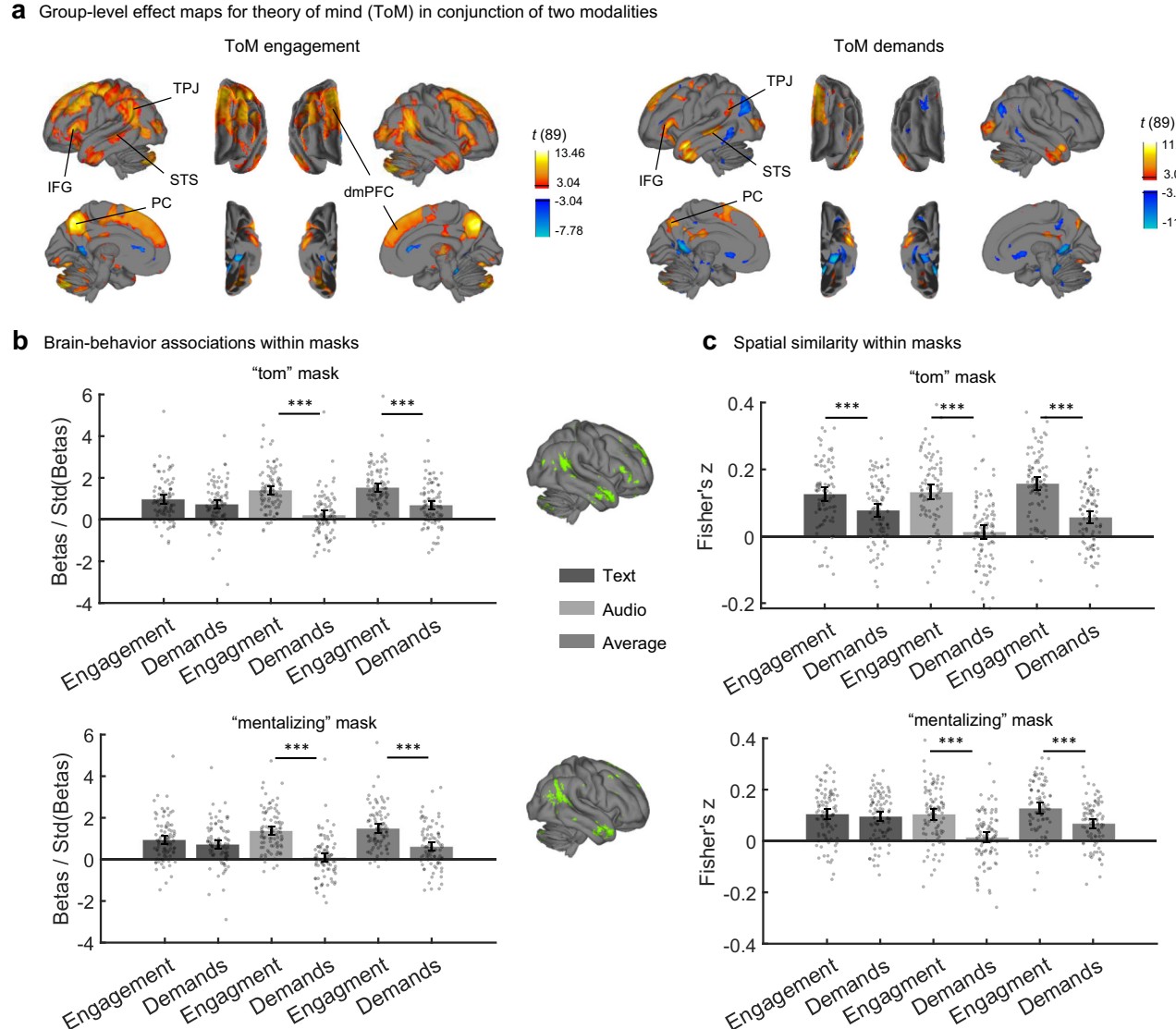

**Fig. 4 | Neural correlates of ToM engagement and ToM demands. a** Group-level effect map of ToM as the medians of online participants' ratings of ToM engagement and the annotation of ToM demands, thresholded at FDR $q < 0.01$. For simplicity, we display only the conjunction maps across modalities (for modality-specific maps, see Fig. S3). The projection from volumetric to surface space and abbreviations are as in Fig. 3a. **b** Group-level comparisons of the effect sizes (Cohen's $d$, mean betas divided by the standard deviation across participants) for ToM engagement ("Engagement") and ToM demands ("Demands") in two modalities within two brain masks. Bar heights indicate group-level effect sizes (Cohen's $d$) and error bars indicate 95% confidence intervals, centered around group means. Points show the mean across voxels for each individual participant. Portions of the masks on the right hemisphere shown in green on the surface plots. All $p$ values were estimated two-sided from 10,000 bootstrap samples that compared two effect sizes within participants ($N = 90$). The $p$ values for the two comparisons in the Text modality were 0.15 and 0.14, respectively, while all other $p$ values for the rest of the comparisons were below 2e-4 (the precision limit of the bootstrap test).
**c** Group-level comparisons of spatial pattern correlations between beta maps and Neurosynth maps within each mask. Bar heights indicate means of Fisher $z$ values (transformed from Pearson's correlation coefficients), and error bars indicate 95% confidence intervals, centered around group means. Scatters indicate the mean $z$ values of each participant. All comparisons were two-sided repeated-sample $t$-tests ($N = 90$ participants), and significance levels were corrected by Bonferroni correction (six comparisons). The raw $p$ values are, from left to right, up to bottom: 1.35e-4, 1.05e-13, 5.25e-13, 0.47, 1.26e-9, 1.6365e-13. In both (**b**, **c**), $^{***}p < 0.001$. std: standard deviation. This figure shows that ToM engagement was more strongly associated with neural activity in canonical ToM regions than ToM demands, and that its spatial association patterns more closely matched meta-analytic ToM maps.

social interactions and/or ToM. These features included intentional actions, all actions, and biological motion of characters; the presence of multiple people; indoor and outdoor scenes; valence of sentences (quantified as sentiment polarity); and reading ease, sentence complexity, part-of-speech, and semantic frames. Most features were unrelated to ratings of social interactions and ToM (absolute Pearson's $r$s < 0.1, $r^2$ < 0.01; Tables S2 and S4) and had distinct neural correlates. For example, neither ratings of actions nor action verbs correlated with social interaction ratings, and their neural correlates showed different patterns from those of social interactions (Fig. S6),

suggesting a separation between social interaction processing and action understanding. Other features that did correlate with social interaction and ToM ratings (e.g., the presence of multiple people) could not explain away the effects of social interactions or ToM (see Supplementary Results for details). Additionally, in predefined brain masks, none of the features were associated with neural activity more strongly than social interactions or ToM (Table S1). These results indicate that the neural correlates of social interactions and ToM are robust to potential confounds.

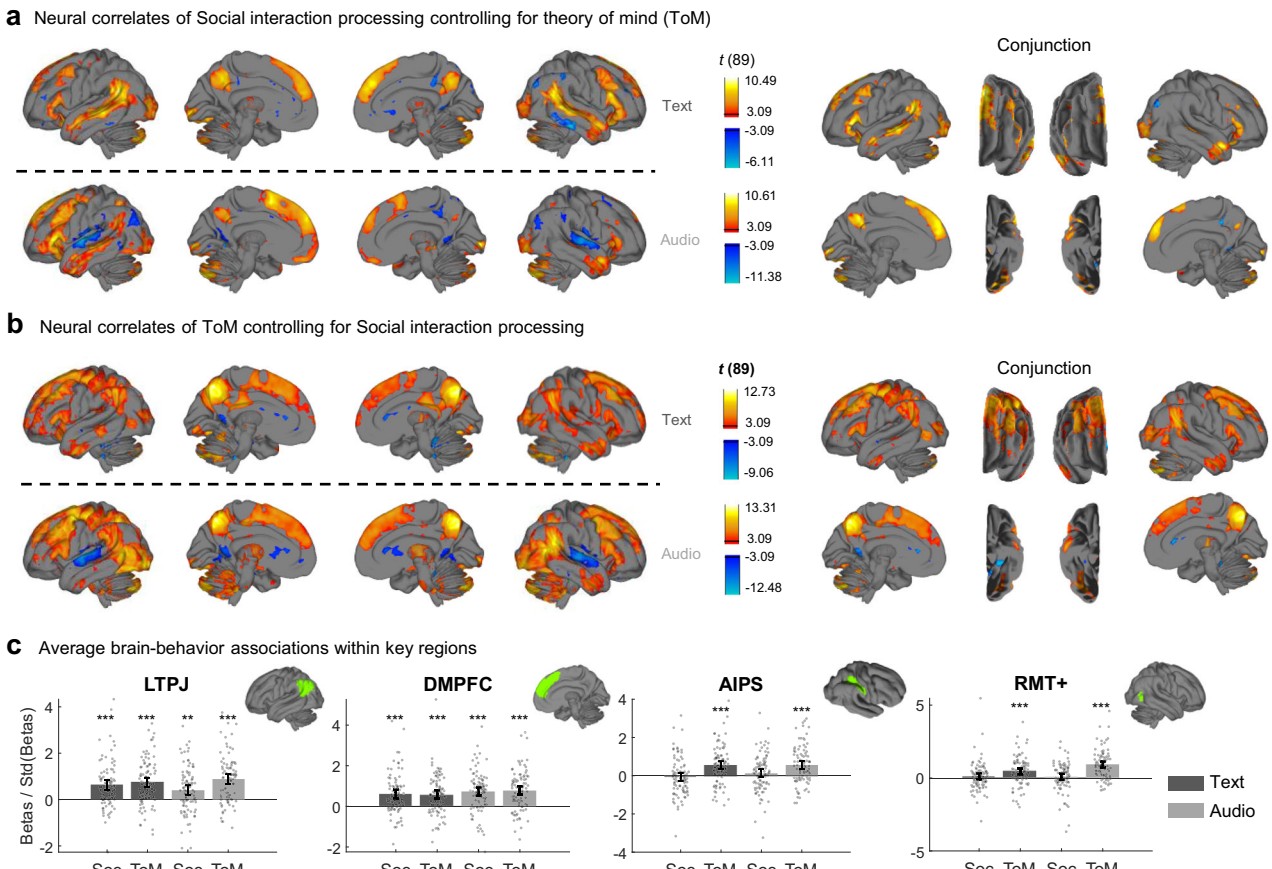

**Fig. 5 | Neural activity associated with social interaction processing and ToM.**
**a**, **b** Group-level effect maps of the medians of social interaction and ToM ratings, each controlling for the other, thresholded at FDR *q* < 0.01. The volumetric-to-surface projection method is the same as in Figs. 3a and 4a. Surfaces on the left show neural correlates for the Text and Audio modalities separately, whereas surfaces on the right show the conjunction across modalities. **c** Brain-behavior associations in four regions of interest (*N* = 90 participants for each bar). Bar heights indicate group-level effect sizes (Cohen's *d*), error bars indicate 95% confidence intervals centered around group means, and shaded scatters indicate estimates for each participant. Each *p* value was obtained from a two-sided one-sample *t*-test against zero. All raw *p* values from left to right: 5.96e-8, 3.45e-10, 1.92e-4, 6.90e-13, 1.19e-7, 4.17e-7, 8.23e-10, 8.86e-11, 0.53, 1.31e-6, 0.22, 8.34e-7, 0.30, 2.83e-5, 0.47, 1.45e-13. ***\*\*\****p* < 0.001; ****\*\****p* < 0.01 (significance levels were adjusted by Bonferroni corrections of 16 comparisons). std standard deviation, LTPJ left temporoparietal junction, DMPFC dorsomedial prefrontal cortex, AIPS anterior inferior parietal sulcus, RMT+ right middle temporal and medial superior temporal area, Soc social interactions. This figure shows that social interaction processing and ToM had overlapping neural correlates in many social cognitive regions, whereas ToM uniquely activated some regions, such as AIPS and RMT+.

The aforementioned results implied that there are distinct neural correlates of social interaction processing and ToM, but could not provide evidence for which voxels or regions have neural activity correlated with only social interaction processing but not ToM, and vice versa. For this, we calculated BF for each voxel and region of interest to quantify the evidence for the alternative and null hypotheses, or presence and absence of an effect. For a voxel or region, if we had significant evidence for the alternative hypothesis for both ToM and social interactions, we classified it as Both; if we had significant evidence for the alternative hypothesis for ToM and the null hypothesis for social interactions, we classified it as ToM only; in the opposite case, we classified it as Social interactions only. We chose 5 as a threshold for the alternative hypothesis and 1/5 for the null hypothesis to obtain moderate to strong evidence, given our sample size[56].

As seen in Fig. 6a, across the two modalities, both social interactions and ToM correlated with neural activity in bilateral PC, dmPFC, IFG, left STS, and left TPJ. In the Text modality, social interactions showed unique correlations with activity in bilateral STS, left visual cortex, and right ventromedial prefrontal cortex (vmPFC); in the Audio modality, left STS and bilateral vmPFC. However, across modalities, social interactions showed only limited unique correlations in small parts of left STS and right vmPFC (0.08% of all voxels). By contrast, ToM

ToM showed cross-modal unique correlations with neural activity in bilateral aIPS, right MT+, right TPJ, and right SMA (1.65% of all voxels). We calculated the mean betas and BF in the same eight selected regions as for the beta maps and found further evidence that there were regions responding to both and regions responding to ToM only consistently across modalities, but not for Social interactions only (Figs. 6b and S5). Across two modalities, activity in about 6.05% of all voxels was correlated with both social interactions and ToM, and those voxels overlapped most strongly with the Default Mode Network (68.1% of those voxels). ToM-only voxels overlapped most strongly with the Frontoparietal Network (27.2%) and Dorsal Attention Networks[51] (21.63%). Social-interaction-only voxels overlapped most strongly with the Default Mode Network (88.3%). Moreover, neural activity in the ToM-only voxels was not significantly associated with the majority of the other features tested across modalities (all absolute mean effect sizes < 0.31, all *p*s > 0.05, all 95% CIs included zero; Fig. S14 and Table S5), and the associations were significantly smaller than that of ToM (all absolute effect size differences ≥ 0.50, all but one *p*s < 0.05, all but one 95% CIs were above zero; Table S5). These results provide strong evidence that the common neural correlates of conceptual processing of social interactions across modalities are a subset of those of ToM during naturalistic experience. Several brain regions are

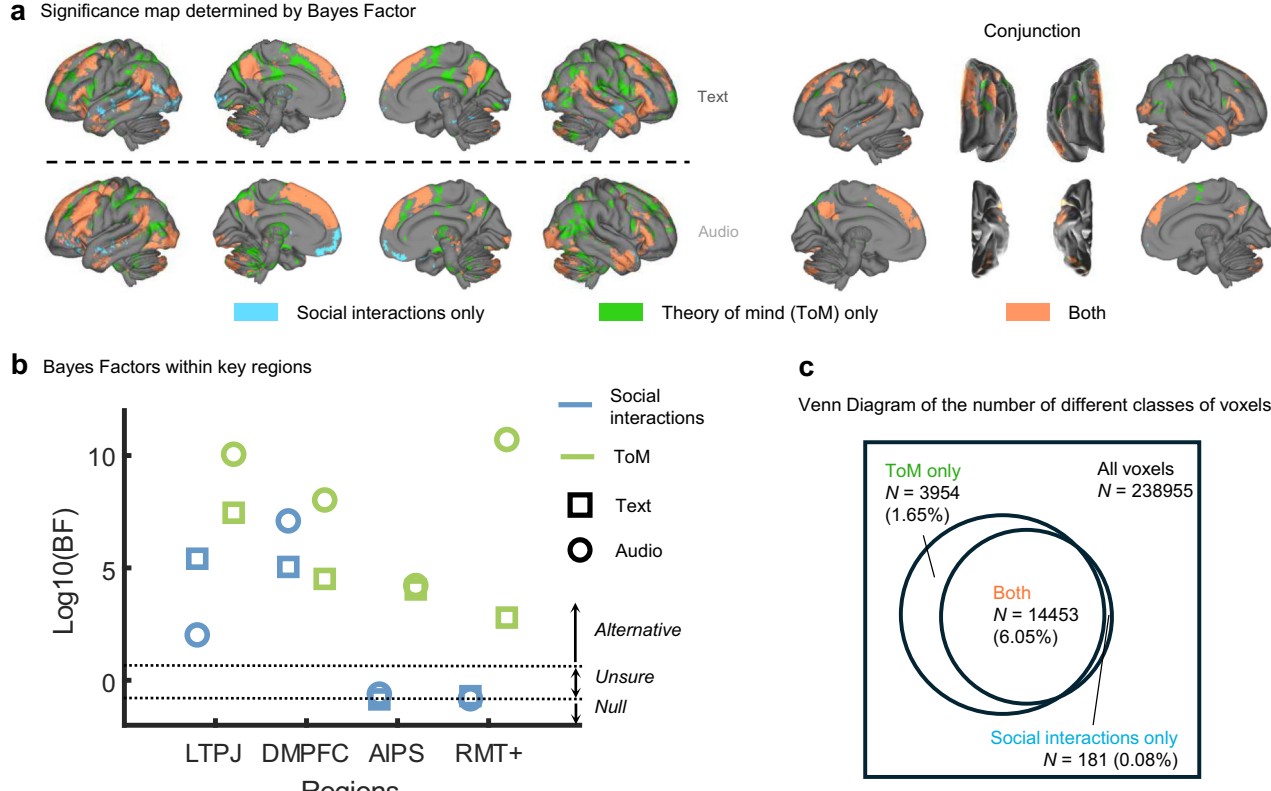

**Fig. 6 | Classifications of voxels/brain regions by Bayes Factors (BF). a** Surface projection of each class of voxels (Social interactions only, ToM only, or Both). The surface space is the same as that in previous figures, except that the interpolation method here is nearest neighbor. Separate results for text and audio are shown on the left, and consistent effects across text and audio ("Conjunction") are shown on the right. **b** Group-level BFs for the same four regions shown in Fig. 5c (using the same abbreviations). The thresholds applied to BFs are shown by the horizontal black dashed line, and the corresponding conclusions in each of the three BF value ranges are indicated by italic labels on the right. For example, if a group effect (point marked with circle or square) has a BF greater than 5 (log10(BF) > 0.70),

there is sufficient evidence for the alternative hypothesis that the effect differs significantly from zero. **c** A Venn diagram demonstrating the proportions of voxels that belonged to each class in the conjunction map shown in Panel a. The percentages in parentheses refer to the number of voxels in one class divided by the total number of voxels in the brain. This figure shows that both social interaction processing and ToM had unique neural correlates, but the unique correlates of social interaction processing rarely overlapped across modalities. Some brain regions showed responses to both social interactions and ToM across modalities, and some showed unique responses to ToM.

only associated with ToM but not with social interactions or other features.

To infer the functional roles of common regions activated by both social interactions and ToM, Social-interactions-only regions, and ToM-only regions, we investigated their spatial correlations with Neurosynth meta-analytic term maps[52] and topic maps[57]. The spatial pattern of voxels activated by both social interaction processing and ToM correlated most strongly with term maps of "person", "social", and "mental", and topic maps of Empathy and interaction and Motor control. Social-interactions-only voxel patterns correlated most strongly with term maps of "sentences", "comprehension", and "text", and topic maps of Trauma and stress and Language comprehension. Lastly, ToM-only voxel patterns correlated most strongly with term maps of "action", "execution", and "hand", and topic maps of Body perception and Motion perception.

## Discussion

Using a combination of normative subjective ratings and neuroimaging during the experience of written narratives, we demonstrated that the neural correlates of processing social interactions and using ToM overlapped in brain regions canonically associated with ToM, including TPJ, dmPFC, PC, and STS. This strong overlap occurred even though the time courses for social interaction processing and ToM were distinct and only weakly correlated. These findings imply that

canonical ToM regions were activated by the mere presence of social interactions in narratives, without explicit engagement of ToM, and that the same areas were further activated when participants engaged in reflective ToM. Distinct areas for each were found as well; in particular, areas in frontoparietal, occipitotemporal, and supplementary motor cortices were selectively involved in ToM and not in social interaction processing. In addition, the associations between social interaction processing/ToM and brain activity were not explained by recognition of actions, biological motion, indoor versus outdoor scenes, valence of narratives (sentiment), or linguistic features, and they remained after controlling for the presence of multiple people in a scene, a driver of social interaction processing. Finally, correlating fMRI activity with moment-by-moment self-report during naturalistic stimuli was a valid and fruitful way to study the neural correlates of ToM, because self-reported ToM was more sensitive than the annotation method widely adopted by previous studies[36–38,44,45].

In the sections below, we compare our results with previous findings and seek an integrated view of the neural correlates of social interaction processing and ToM. First, we argue that many social interaction processing regions are involved in both perceptual and conceptual processing. There is rich evidence for the neural correlates of the visual perception of social interactions and for specific visual features embedded in social interactions, such as action directions[24] and motion information[22]. Only recently have researchers noted the

contribution of heteromodal processing to neural activity underlying social interaction processing, particularly in STS[6,29]. We extended this line of work by showing that verbal descriptions of social interactions engaged brain regions similar to those activated by visual and audio-visual presentations, both in STS and beyond. For example, these included regions most often implicated in visual perception, such as LOC and ITC, even in the audio modality when there were no visual stimuli at all (Fig. 3a). These regions overlapped with those found by social animations[10] and were thus likely involved in modality-independent conceptual processing of social interactions, broadly consistent with predictive processing accounts of vision[19].

Second, our results show that social interaction processing covaries with ToM and activates ToM regions, further implying that social interaction processing and ToM share pre-reflective inferential processes. By correlating fMRI time courses with normative self-reports, we quantified the moment-by-moment recognition of social interactions and use of ToM. We found that people's tendency to use ToM was positively correlated with their judgments of social interactions to a small to moderate extent. This finding supports the idea that recognizing social interactions motivates people to use ToM to some degree[39–41], while also indicating that the two processes are distinct and separable. Additionally, almost all brain regions correlating with social interactions across modalities also responded to ToM above and beyond the effects of recognizing social interactions. Because we included both social interactions and ToM in the same model and they were behaviorally separable, it is unlikely that the neural correlates of social interaction processing could be fully explained by ToM. Instead, the overlapping neural correlates likely implement computations shared between social interaction processing and ToM.

One possible shared computation is pre-reflective inferences on social cues. Most of the common neural correlates of social interaction processing and ToM were associated with the Default Mode Network (DMN)[51], which primarily consists of transmodal regions[58,59] that are functionally distant from unimodal cortices and encode higher-level conceptual information during narrative experiences[60–62]. Moreover, these regions were significantly activated by the presence of multiple people (see Supplementary Results), a key precursor to recognizing social interactions, and their spatial patterns were most similar to the "person" term map among 525 Neurosynth term maps. The most parsimonious explanation is that these regions respond to features simpler than and preceding recognition of social interactions and mental states, or social primitives[9] in the narratives, even when participants do not explicitly report mentalizing. This is congruent with the idea that the more demanding ToM might have recycled existing neural infrastructure for social interaction processing (especially the DMN) during evolution[7] and suggests the existence of non-deliberative or pre-reflective processing in these regions. Thus, the shared neural correlates between social interaction processing and ToM may serve the automatic engagement of modality-independent, pre-reflective inferences on social primitives[2].

Moreover, these inferences may function to simulate the social world[63] and prepare individuals to act in it whenever there is a need to understand an agent outside the self, here, and now. These needs include understanding social interactions and others' mental states, as well as disambiguating animations[64] or recollecting autobiographical memories[65]. Another potential function of these inferences is to support automatic ToM, if they are unconscious and contain enough information to solve ToM tasks[34,35]. However, our paradigms did not allow us to directly quantify automatic ToM and link it to neural activity. Future studies will need more elaborate, controlled experiments to tease automatic ToM apart from controlled ToM and investigate whether any brain regions are activated only by automatic ToM.

Intriguingly, the common neural correlates of social interaction processing and ToM also included parts of the early visual cortex (V3 and V4) and ITC. This finding suggests that the ventral visual stream is involved in processing amodal, conceptual information in social narratives. The specific information represented there could be visual or mental imagery of story content[66,67] or story-elicited emotional concepts[68]. Future studies are needed to investigate what information causally activates those regions during narrative processing.

Lastly, we found that self-report ToM uniquely activates regions beyond social interactions and controlled ToM tasks. These regions were widespread, appearing mainly in bilateral aIPS, right LOTC or MT +, right TPJ, and right SMA. Except for the right TPJ, these regions were not only insensitive to social interactions but also rarely reported in controlled ToM tasks[13,14]. However, some of them have shown activation in tasks that involve social motion ("Social Animations"[5]). The ToM-only regions overlapped mostly with the Frontoparietal Network and were most similar to the Neurosynth-based term maps of "action" and topic maps of Body perception and Motion perception[69,70]. Given that intentional actions or biological motion, regardless of sociality, were uncorrelated with ToM and did not significantly activate the ToM-only regions (see Supplementary Results), it is possible that these regions are specifically sensitive to action cues in social contexts.

Some limitations of the current study are worth noting. First, most of our results were based on group-level statistical inferences. Although this provided insights into on-average effects, important individual differences were likely omitted, especially considering the idiosyncratic nature of ToM. More work is needed to develop new ways to measure ToM engagement and relate it to neural activity at the individual level. Second, given the limited spatial resolution of fMRI, we may have missed some larger clusters that respond only to social interactions across modalities. Third, to allow more objective annotations for ToM demands and easier self-report of ToM engagement, we adopted a more generic definition of ToM as the mental process of thinking about other people's mental states. Although it captures the core definition of ToM, it cannot reveal different sub-processes involved in ToM[71] or the specific ways people infer others' mental states[12]. It is also possible that our ToM measures captured other mental processes distinct from those typically engaged during reflective ToM tasks.

In conclusion, the current study shows that the conceptual processing of social interactions relies on a subset of the brain regions that respond to ToM in narratives, while ToM has unique neural correlates beyond canonical ToM regions. The findings imply that both recognizing social interactions and using ToM involve pre-reflective inferences (e.g., pattern recognition processes) that rely on regions overlapping mostly with the DMN. In addition, self-reported ToM may engage processes in frontoparietal systems, such as processing social action cues. Future research is needed to test the reliability and generalizability of these findings and to refine the theoretical accounts for them.

## Methods

### Behavioral ratings of social interactions and ToM

**Participants.** A total of 268 participants took part in our online experiment from Prolific.com. The institutional review board (IRB) of Dartmouth College approved to conduct the study (CPHS STUDY00032930), and all participants provided informed consent. Each participant was compensated $8 USD after they completed the study. Of these, 32 participants failed more than half of the attention check questions, two participants provided the same ratings throughout the experiment, and three participants did not have their data saved. The remaining 231 participants with valid data (104 females, 127 males) are all healthy adults (19–45 years old, $M = 33.65$, $SD = 6.51$) whose native language is English and who have no literacy difficulties. The sex of participants was determined based on self-report and was not considered in the study design. All participants have an approval rate (proportion of valid data from studies they have taken part in) of over 95% on Prolific.

**Table 1 | Properties of the narrative stimuli**

| Run # | Narrative # | Modality | Duration (mm:ss) | Number of TRs | Number of words |
|---|---|---|---|---|---|
| 1 | 7 | Audio | 2:01 | 263 | 429 |
| 1 | 8 | Audio | 1:36 | 209 | 314 |
| 2 | 5 | Audio | 2:30 | 326 | 511 |
| 2 | 6 | Audio | 2:04 | 269 | 447 |
| 3 | 3 | Text | 3:11 | 415 | 573 |
| 3 | 4 | Text | 3:29 | 454 | 626 |
| 4 | 1 | Text | 2:43 | 355 | 490 |
| 4 | 2 | Text | 2:40 | 348 | 480 |

The table is sorted by run number, followed by narrative number. The sequence of the two narratives within a run was counterbalanced across participants.

**Stimuli**. The main stimuli are eight short narratives (stories) in the third-person perspective. Each narrative includes a single protagonist who moves along a single storyline composed of nine different situations, which are sampled from Polti's 36 dramatic situations[72] and characterized by interpersonal relationships and actions. During the experiment, each situation or part was presented in one experimental trial. The first four narratives were presented as audios, where an AI-generated adult male voice read out the narratives. The second four were presented as white texts on a black full-screen background at a constant speed of three words per second; the number of words on each screen was 10 or the number of remaining words in a part, whichever was smaller (Fig. 1a). Table 1 below provides a more detailed description of the duration and length of each narrative, the presenting modality, and the duration in the unit of volumes (TRs) in the neuroimaging study.

Importantly, the variety of situations and transitions between them make each narrative contain rich descriptions of both the activity of a single character and the joint activity of multiple characters, allowing for a contrast over social interactions. Similarly, some parts of the narratives contain dense descriptions of characters' mental states, for example:

She was furious that she did not know what was going on...

while others do not, for example:

Amy and Linda continued to be best friends...

The narratives are thus capable of inducing different levels of ToM engagement at different times during the narratives.

**Experimental procedures**. Potential participants open the study link from Prolific.com and will be directed to the webpage-based experiment hosted on Pavlovia.org. There, they learn more about the experiment and provide their consent to participate before starting the experiment.

The experiment consists of four runs. Between two runs, participants could take a short break of no more than 1 min. In each run, two narratives are played as texts or audios (Table 1). Because each narrative is presented in nine experimental trials, there are 18 trials in each run. At the start of the first and third runs, there is an additional practice trial for participants to get familiar with the experimental interface and ratings. One similar but irrelevant narrative part is used in the practice trials.

Each trial starts with a fixation period with pseudo-randomized durations between 2 and 8 s. Then a narrative part is presented, while participants need to give their ratings moment by moment by moving their mouse on top of a continuous visual analog scale with two anchors at the ends and one anchor at the midpoint. The rating question is randomly chosen for each participant. About half of the participants have answered the social interaction question:

Are there social interactions? (How much do you think there are social interactions in the narrative at this moment?)

with three anchors being "Not at all", "Partially", and "Definitely yes". The other half of the participants have answered the ToM question:

Thinking about mental states? (How much are you thinking about the mental states of characters in the narrative at this moment?)

with three anchors being "Not at all", "Half the time", and "All the time". Questions in the bracket are only presented in practice trials. At the end of each trial, participants are asked to give an intermittent rating about the previous trial:

On average, how much [were you thinking about the mental states of characters]/[do you think there were social interactions] during the last part of the narrative?

The anchors are the same as the previous ratings (Fig. S1). The intermittent ratings have not been analyzed or reported in this article.

The sequence of the two narratives within each run is counterbalanced across participants. To check whether participants were paying attention, in one random trial per run, a surprise question would follow the intermittent rating and ask participants to click on a specific location of the scale.

**Data collection**. All ratings were recorded as a float number between 0 and 100, linearly mapping onto the positions of the visual analog scale. The moment-by-moment ratings were sampled every 83 ms or every cycle of the participants' monitor refreshing, whichever was longer. Because we anticipated a delay in participants' ratings as compared to the narrative flow (e.g., a rating of yes to social interactions might happen hundreds of milliseconds after social interactions appeared), we continued collecting ratings for another 1.5 s after each narrative part ended.

As stated in the Results section, researchers annotated ToM demands in the narratives offline. ToM demands reflect the moment-to-moment needs for participants to use ToM and are defined by whether there is explicit mention about mental states of characters in the narratives. Before any annotation, the narratives were cut into sentence parts, the smallest chunk in a sentence that contains one or a sequence of predicates and their associated object[53] (see Supplementary materials for a full list of sentence parts). The sentence parts were presented in their original orders for researchers to read at their own pace. They provided yes/no (1/0) labels to each sentence part about whether it contains ToM demands. When there was a conflict of annotations (when two raters provided the same label while the other two opposed), a consensus label was generated after further discussion.

**Quantification and statistical analysis**. To ensure data quality, we excluded ratings of a participant from further analysis if they failed more than two out of the four attention check questions ($n = 32$) or kept providing the same rating throughout the experiment ($n = 2$). During the first few seconds of each trial, participants might not have started engaging in the narrative and providing meaningful ratings. Thus, we set as missing values all the moment-by-moment ratings sampled before the first move of the mouse in each trial. Next, to aggregate and compare moment-by-moment ratings across

participants, we down-sampled them to 230 ms per sample (chosen as half of the resolution of the BOLD data in the neuroimaging study) by linear interpolation. In generating a group-level summary statistic that was later applied to neuroimaging data, we excluded data from participants whose social interaction ratings had a correlation with the medians of all other participants' ratings lower than 0.3 ($n = 7$), as well as those whose ToM ratings had a correlation lower than 0.2 ($n = 14$).

To estimate the extent to which online participants had similar ratings for social interactions and ToM, two metrics were calculated. First, we randomly divided participants into two equal-sized groups and correlated the mean ratings of the two groups. This split-half correlation value was corrected by the Spearman–Brown formula[48,49]. We repeated the sampling and calculation 2000 times randomly to estimate the mean and CI of the correlation coefficients. Second, we calculated the pairwise correlation between participants' ratings and found the range, mean, and median of the correlations. The latter metric is a more conservative estimate of how similar participants were in their ratings.

We also investigated the extent to which researchers' annotations of ToM demands agreed with online participants' ratings of ToM engagement. To do that, we resampled the annotations to 230 ms per sample by linear interpolation. We then extracted the medians of online participants' ratings as a group-level summary and calculated their correlation with researchers' annotations. The statistical significance of the correlation was estimated by 10000 bootstrap samples with phase randomization[73].

## Neuroimaging study on social interaction processing and ToM

**Participants.** Ninety-seven participants took part in an fMRI experiment. The IRB of Dartmouth College approved to conduct the study (CPHS STUDY00031937), and all participants provided written consent. Data from one participant were missing due to technical issues, and data from another six participants were excluded from further analysis because of failures in fieldmap collections, which are essential for the susceptibility distortion correction step in fMRI preprocessing. The remaining 90 participants with valid data (55 females, 34 males, 1 other) are healthy adults (18–45-years-old, $M = 24.72$, SD = 5.57) without any current physical or mental disorders. The sex of participants was determined based on self-report and was not considered in the study design. All participants are native English speakers or have comparable fluency in English, and have normal or corrected-to-normal vision.

**Stimuli and experimental procedures.** The neuroimaging study is part of a larger project about the neural correlates of various cognitive and affective processes[74]. The project was conducted at the Dartmouth Brain Imaging Center and required participants to pay for four separate visits on different days. Participants were monetarily compensated after the entire project. The data analyzed here came from the "narratives" task, which was performed on the second visit of every participant. At the beginning of their second visit, all participants provided their written consent to continue participation in the study. After that, they practiced every task they would perform that day on a laptop under the experimenters' instructions. All instructions were previously scripted and read out loud to each participant. When participants had no further questions about the tasks, they would lie in the scanner and begin the formal tasks.

The run structure and narratives played in each run were the same as in the behavioral study (Table 1). The main difference between the two studies was in the trial structure: fMRI participants did not provide any ratings when the narratives were presented; after one narrative part finished, they were asked to provide two ratings about their feelings and expectations. The ratings were not analyzed or reported in the current article (Fig. S1).

**Data collection.** Structural and functional MRI data were acquired on a 3T Siemens MAGNETOM Prisma MRI scanner with 32-channel parallel imaging. Structural images were acquired using high-resolution T1 spoiled gradient recall images. Functional images were acquired with a multiband EPI sequence (repetition time (TR) = 460 ms, echo time (TE) = 27.2 ms, field of view = 220 mm, multiband acceleration factor = 8, flip angle = 44°, 64 × 64 matrix, 2.7 × 2.7 × 2.7 mm voxels, 56 interleaved ascending slices, phase encoding posterior to anterior). Stimuli were programmed and presented by Psychtoolbox (MATLAB, MathWorks) on a Linux laptop.

## Quantification and statistical analysis

**MRI data preprocessing.** Preprocessing was performed using fMRI-Prep 21.0.2[75] based on Nipype 1.6.1[76]. To correct for magnetic field inhomogeneity, a B0-nonuniformity map (fieldmap) was estimated based on two echo-planar imaging (EPI) references with `topup`[77] (FSL 6.0.5.1:57b01774).

For each participant, one T1-weighted (T1w) image was corrected for intensity non-uniformity with `N4BiasFieldCorrection`[78], distributed with ANTs 2.3.3 (RRID:SCR_004757)[79], and used as T1w-reference throughout the workflow. The T1w-reference was then skull-stripped with a `Nipype` implementation of the `antsBrainExtraction.sh` workflow using OASIS30ANTs as the target template. Brain tissue segmentation of cerebrospinal fluid (CSF), white-matter (WM) and gray-matter (GM) was performed on the brain-extracted T1w using `fast`[80] (FSL 6.0.5.1:57b01774, RRID:SCR_002823). Brain surfaces were reconstructed by `recon-all`[81] (FreeSurfer 6.0.1,RRID:SCR_001847). The brain mask estimated previously was refined with a custom variation of the method to reconcile ANTs-derived and FreeSurfer-derived segmentations of the cortical gray-matter of Mindboggle[82] (RRID:SCR_002438). Volume-based spatial normalization was performed through nonlinear registration with `antsRegistration` (ANTs 2.3.3), using brain-extracted versions of both the T1w reference and the T1w template. Two templates were selected for spatial normalization: ICBM 152 Nonlinear Asymmetrical template version 2009c (MNI152NLin2009cAsym, RRID:SCR_008796; TemplateFlow ID: MNI152NLin2009cAsym) and FSL's MNI ICBM 152 non-linear 6th Generation Asymmetric Average Brain Stereotaxic Registration Model (RRID:SCR_002823; TemplateFlow ID: MNI152NLin6Asym).

For each of the four BOLD runs per subject, the following preprocessing was performed. First, a reference volume and its skull-stripped version were generated by aligning and averaging 1 single-band reference (SBRefs) using a custom method of fMRIPrep. Head-motion parameters with respect to the BOLD reference (transformation matrices, and six corresponding rotation and translation parameters) are estimated before any spatiotemporal filtering using `mcflirt`[83] (FSL 6.0.5.1:57b01774). The estimated fieldmap was then aligned with rigid registration to the target EPI reference run. The field coefficients were mapped onto the reference EPI using the transform. The BOLD reference was then co-registered to the T1w reference using `bbregister` (FreeSurfer), which implements boundary-based registration[84], generating a preprocessed BOLD run in MNI152NLin2009cAsym space. Co-registration was configured with nine degrees of freedom to account for distortions remaining in the BOLD reference. Several confounding time series were calculated based on the preprocessed BOLD: FD, DVARS, and three region-wise global signals. FD and DVARS were calculated for each functional run, both using their implementations in `Nipype`, following the definitions by Power and colleagues[85]. The three global signals were extracted within the CSF, the WM, and the whole-brain masks. The six head-motion estimates were expanded with the inclusion of temporal derivatives and quadratic terms for each[86]. Gridded (volumetric) resampling was performed using `antsApplyTransforms` (ANTs), configured with

Lanczos interpolation to minimize the smoothing effects of other kernels[87]. Lastly, the functional data were spatially smoothed with a 6 mm full-width-at-half-maximum (FWHM) Gaussian kernel.

**Voxel-wise GLMs.** To find the neural correlates of variables of interest (social interactions and ToM), we applied a hierarchical random effect model approach using two-stage summary statistics. Specifically, we applied a custom probability brain mask (https://github.com/canlab/CanlabCore/blob/master/CanlabCore/canlab_canonical_brains/Canonical_brains_surfaces/brainmask.nii) thresholded at a probability of 0.5 to the preprocessed BOLD images and resampled the original voxels to form new $2 \times 2 \times 2$ mm voxels by spline interpolation. Then, in each voxel, we fit a GLM for each participant using the ordinary least squares method. We aggregated the regression coefficients in each voxel across participants and performed one-sample $t$-tests and robust regression[88] on single coefficients or planned contrasts over several coefficients. Results from $t$-tests and robust regression were almost identical, and we only reported those from $t$-tests. To find labels for the significant brain regions on the cortex, we overlaid the thresholded $t$ maps on a multi-modal parcellation of the human cerebral cortex[50] and calculated the percentage of significant voxels within each parcel. Parcels with more than 25% of significant voxels were treated as significant, and adjacent parcels were grouped into larger regions with more commonly used names (such as TPJ instead of PGi, PGs, and PFm).

Variables of interest came from two different sources: online participants' moment-by-moment ratings and researchers' offline annotations. The medians of online participants' ratings were linearly projected from the range [0, 100] to [−1, 1] before being entered as regressors in the GLM. To ensure the medians were valid and meaningful, for each trial, we excluded data from participants who had less than 40% of non-missing moment-by-moment ratings. We also excluded the time points when less than 50% of the participants had non-missing rating values by adding indicator functions to the GLM whose values were 1 at the excluded time points and 0 at all other time points. Researchers' annotations of ToM demands were coded as two separate binary regressors with values 1 or 0. One regressor modeled the sentence parts where there were ToM demands, and the other modeled where there were no ToM demands; for a sentence part with ToM demands, the first regressor had a value of 1 and the second a value of 0. Thus, a contrast between the estimates of the two regressors revealed the effects of ToM demands.

To control for the effect of answering questions on neural activity, we added a binary regressor whose values were 1 when participants were giving ratings at the end of each trial and 0 otherwise. This rating regressor, together with all variables of interest, was convolved with the canonical hemodynamic response function[89]. This made the fixation periods the only implicit baseline of the BOLD signals.

Some further data denoising was performed in the GLM. First, we high-pass-filtered the functional data at 1/128 Hz by adding a series of discrete cosine functions to each model. Second, we added 24 head-motion-related parameters to account for the effects of head motions, including six head motion parameters (x, y, and z translations, and pitch, yaw, and roll), their temporal derivatives, their squares, and the squares of the derivatives. Third, we added the mean signal in the cerebrospinal fluid as an approximation of physiological artifacts. Lastly, we identified outlier brain volumes by searching for the volumes whose average signals or framewise displacements (FDs) were outside three standard deviations from the mean of each run. We excluded those volumes by entering each of them as an indicator function whose values were 1 at the time point of the outlier and 0 at all other time points.

Because the functional signals at the start of each run may be unstable and deviate from the rest of the run, we discarded the first 6 volumes in each run when there were no stimuli on the screen. The rest of each run's functional data were rescaled to have a grand mean of 100 (across all volumes and voxels) to account for differences in signal baselines across runs and participants[90]. Functional data were concatenated across all four runs, as were the design matrices. Variables of interest, as well as the rating regressor, were shared across runs, such that only one beta value was estimated for each variable of interest in each voxel. All other variables, including the intercept, were modeled separately for each run.

**Mask-based analysis.** We performed analysis within several brain masks defined by results from previous studies or by well-established brain parcellations. The first set of masks was defined by term-based association maps from Neurosynth.org[52]. The $z$-scores from a two-way ANOVA in those maps quantify the possibility of each voxel being positively activated in a paper that includes a specific term. The original maps were thresholded at FDR $q < 0.01$, and we further thresholded them by $z \geq 3$ to reduce false positive voxels and make the masks more specific. The terms used in the current study include: "social interaction", "social cognition", "psts", "tom", and "mentalizing".

The second set of brain masks was based on the ToM group maps from a false belief task[55]. The masks include bilateral TPJ (combining the original LTPJ and RTPJ maps), RSTS (the original RSTS map), bilateral PC (the original PC map), and bilateral dmPFC (combining the original DMPFC and MMPFC maps). We did not use the original VMPFC map because no GLM results in the current study showed significant effects in the ventral medial prefrontal cortex.

The third set of masks was from a multimodal parcellation of the cerebral cortex[50] (the Human Connectome Project Multi-Modal Parcellation version 1.0). Each mask was a union of several parcels based on their anatomical positions and functional topography. For a full list of masks used in the current study and how we defined them, see Table 2. Within each mask defined in any of the three ways, we extracted individual-level regression coefficients from the GLM analysis and compared the group mean to zero to investigate whether a variable of interest had consistent neural activations across participants in that mask.

In comparing the explanatory effects of different features across models within the same brain mask (Figs. 4, S4, and S12; and Tables S1 and S5), we used group-level Cohen's $d$ against zero as a scale-free measure of effect sizes. Within each brain mask, a $d$ was calculated as the grand mean regression coefficient (beta) divided by its standard deviation across participants. To test whether the two effect sizes, $d_1$ and $d_2$, were significantly different, we performed a bootstrap analysis with 10,000 samples to estimate a distribution for the mean of $d_1 - d_2$. A $p$ value was calculated as the proportion of bootstrap means that were 0 or of the opposite sign to the observed

**Table 2 | Masks defined on the Human Connectome Project multi-modal parcellation version 1.0[50]**

| Mask | Parcels included | Hemisphere |
|------|------------------|------------|
| LTPJ | PGi, PGs, PFm | Left |
| dmPFC | d32, 8BM, 9m | Bilateral |
| AIPS | PFt, AIP, IP2 | Bilateral |
| RMT+ | MT, MST | Right |
| PC | 7m, v23ab, d23ab, 31pv, 31pd | Bilateral |
| LSTS | STSda, STSdp, STSva, STSvp, TPOJ1, TPOJ2 | Left |
| RTPJ | PGi, PGs, PFm | Right |
| SMA | 6ma, 6mp | Bilateral |

An entry of bilateral in the hemisphere column denotes that the parcels on both hemispheres were concatenated to form a single brain mask for analysis. Left and right denote that parcels on only one hemisphere were included.

mean, multiplied by 2 to approximate a two-tailed test. Significance levels were corrected by the Bonferroni Correction.

**BF analysis.** To identify voxels associated with both social interactions and ToM, and those associated with one but not the other, we took the BF approach to quantify evidence for both alternative and null hypotheses. Specifically, the BF value of a given voxel is proportional to the ratio of the likelihoods under the alternative and null hypotheses (e.g., the neural activity in that voxel is correlated with social interactions in narratives or not). We estimated BFs from $t$-values of one-sample tests, assuming a Jeffreys–Zellner–Siow (JZS) prior, following the formula[56]:

$$BF_{10} = \frac{\int_0^\infty (1 + Ngr^2)^{-1/2} \left(1 + \frac{t^2}{(1 + Ngr^2)(N-1)}\right)^{-N/2} (2\pi)^{-1/2} g^{-3/2} e^{-1/(2g)} dg}{\left(1 + \frac{t^2}{N-1}\right)^{-N/2}}$$

(1)

In the formula, $t$ represents the $t$-statistic, $N$ the sample size, $r$ the scale factor, and $g$ the prior variance of the standardized effect size. We set the scale factor to $\sqrt{2}/2$, assuming a moderate effect a priori[56], to increase the support for alternative hypotheses given the frequently smaller effect size in fMRI univariate analysis[57]. In estimating the likelihood for the alternative hypothesis (the numerator), we marginalized over all possible values of g to incorporate the uncertainty about the variance of the effect size[56].

BFs calculated in this manner are asymmetric for alternative and null hypotheses, because evidence for null hypotheses is bound by $t = 0$. Given the current sample size ($N = 90$), the BF cannot be smaller than 0.12 but can be infinitely large. Thus, we chose a heuristic threshold of 5 for the alternative hypothesis and 1/5 for the null hypothesis to attain moderate to strong evidence in favor of both hypotheses.

After calculating the BF for a voxel under one modality, we classified the voxel into one or none of the three categories (Fig. 6). We only considered positive effects (positive $t$ statistics) as being significantly associated with social interactions or ToM in this analysis. The specific criteria are as follows:

- Both− BF > 5 and $t > 0$ for both social interactions and ToM.
- Social interactions only− BF > 5 and $t > 0$ for social interactions, and BF < 1/5 for ToM.
- ToM only− BF > 5 and $t > 0$ for ToM, and BF < 1/5 for social interactions.

To create the conjunction classification map across modalities (Fig. 6a), we assigned a label to one voxel in the conjunction map if and only if it had the same label under the two modalities. This provides a more conservative and specific estimate of the modality-general neural system responding to social interactions only, ToM only, or both.

### Reporting summary
Further information on research design is available in the Nature Portfolio Reporting Summary linked to this article.

## Data availability
All data reported in the current study are openly accessible. All ratings and annotations of the narratives have been deposited in Zenodo [https://doi.org/10.5281/zenodo.18315909][91]. The raw neuroimaging data (in the Nifti format and BIDS structure) can be found at [https://doi.org/10.18112/openneuro.ds005256.v1.1.0] and have been described extensively in a data paper[74]. The first-level beta images generated from raw neuroimaging data have been deposited in Zenodo [https://doi.org/10.5281/zenodo.18316150][92]. The Neurosynth dataset used in the present study is compiled on GitHub and can be found at https://

github.com/canlab/Neuroimaging_Pattern_Masks/tree/master/Neurosynth_maps. The ToM group maps[55] can be found at https://saxelab.mit.edu/use-our-theory-mind-group-maps/. The multimodal parcellation of the cerebral cortex[50] can be found at https://github.com/canlab/Neuroimaging_Pattern_Masks/tree/master/Atlases_and_parcellations/2016_Glasser_Nature_HumanConnectomeParcellation.

## Code availability
Open-source software used in the current study include CANlab neuroimaging analysis tools (https://github.com/canlab) based on MATLAB R2022b (MathWorks), PsychoJS 2023.2.2 nltools 0.5.0 (https://nltools.org/) based on Python 3.9.13, and fMRIPrep (https://fmriprep.org/en/stable/) version 21.0.2. Online data collection and storage used Pavlovia.org, Prolific.com, and Qualtrics. All custom scripts used to analyze and visualize the data can be found at [https://doi.org/10.5281/zenodo.18315909][91].

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

## Acknowledgements

Thanks to Bogdan Petre, Yaroslav O. Halchenko, David M. Gantz, Sydney L. Shohan, Xinming Xu, Jason A. Davis, Jiahui Guo, Maryam Amini, Bethany J. Hunt, and Eilis I. Murphy for data collection and management. This project was supported by grants NIBIB R01EB026549 (T.D.W. and M.A.L.).

## Author contributions

Z.M. and T.D.W. conceived research questions. Z.M., H.J., P.A.K., M.A.L., and T.D.W. designed experiments. H.J. and P.A.K. prepared experimental materials. H.J. performed the neuroimaging experiment. Z.M. implemented and performed behavioral experiments. Z.M., K.B., and P.S. wrote analysis code and performed analysis. Z.M., H.J., P.A.K., P.S., and T.D.W. interpreted the results. Z.M. wrote the manuscript and supplementary information. Z.M., K.B., and T.D.W. revised the manuscript. T.D.W. and M.A.L. supervised and obtained funding for this work. All authors commented on or edited the manuscript at all stages.

## Competing interests

The authors declare no competing interests.
