## [Transparent Peer Review file · Nature Communications]

Common and distinct neural correlates of social interaction processing and theory of mind in narratives

Corresponding Author: Professor Tor Wager

Version 0:

Reviewer comments:

Reviewer #1

(Remarks to the Author)

The authors have developed a most ingenious paradigm for studying social interactions. In the first stage observers continuously rate visual or verbal presentations of social interactions, either for perception of social interactions or for involvement of ToM. There is a very high degree of consistency in these ratings across observers with clear distinctions between the two kinds of rating. In the second stage volunteers are scanned while exposed to the same presentations. The elicited brain activity can then be correlated with the ratings acquired in the first stage to indicate brain locations showing the same fluctuations. The locations so identified are highly consistent with previous work on social interaction perception and ToM. This paradigm has a great future. I shall now consider some of the highlights and problems of the study

1. Comparison of visual and verbal presentations. The activity elicited by these two kinds of presentation was very similar, suggestion that the mode of presentation is not too critical. Interestingly, verbal presentations elicited activity in early visual areas. Was this a result of visual imagery? Do listeners report this? Another interesting observation was that similarity of verbal and visual presentations was reduced when talking into account ToM. This is consistent with the idea that ToM is a higher cognitive level of more abstract processing.
2. The authors make an interesting and important distinction between engagement and demand of ToM. They show that observers can make this distinction. However, I don't think this design allows brain areas associated with demand to be revealed. A design might be used in which there was a period during which volunteers anticipated having to do a ToM task, but without the actual task. This should elicit activity associated with demand.
3. There is a brief mention of action understanding and the mirror system. To what extent does action understanding overlap with social interaction perception. The authors might say a bit more about this.
4. I was concerned that there was quite a bit of reverse inferencing. For example, in relation the extra activity associated with ToM. Could authors justify this better.
5. The authors suggest that ToM is a system 2 process that is slow and effortful. While social interaction perception is a system 1 process. However, there is a literature, starting with Apperly & Butterfill Psych Rev 2009, suggesting that there is system 1 type version of ToM (see Schneider et al. Cognition, 2017). Do the authors see this process as part of social interaction perception? Or do they need to allow for an automatic version of ToM. I note that in line 556 the authors suggest that ToM 'may involve more automatic processing than was previously thought'. Yet, clearly others were previously thinking this.

Reviewer #2

(Remarks to the Author)

Miao et al. present an fMRI study investigating responses to social interactions and theory of mind (ToM) during narrative comprehension. They use audio and text-based stories combined with moment-to-moment behavioral ratings of social

interaction and ToM from behavioral participants. In a separate group of participants, they measure fMRI responses to the same narratives, and evaluate how neural responses relate to interaction and ToM content. They find responses to both social interaction and ToM content in a similar set of brain regions, which have been previously implicated in social cognition. Overall, I found the study interesting and methodologically sound. I have some reservations about the strength of the conclusions given the possibility of confounding factors driving responses to features in naturalistic stimuli, as described in the points below.

MAJOR POINTS

1. The core claims made by the paper rely on relationships between fMRI responses and abstract features (social interactions, ToM) in complex, naturalistic story stimuli. Given the complex nature of the stimulus, there are many potential confounding stimulus features that could be related to the features of interest: for instance, the presence of people at all, the number of people in a scene, the presence of body movements or intentional action, etc. These various confounding factors are not measured or modeled in the analysis. It is therefore not clear that the effects ascribed to social interaction and ToM in this paper don't result from an influence of confounding factors. The authors could make a stronger case for effects specific to social interaction if they were to quantify multiple features relevant to social cognition, and assess their independent effects on fMRI responses
2. The authors frame their results in terms of "social interaction perception." In the literature, this term has typically been used to describe inferences about social interaction directly from visual or auditory cues. In the current manuscript, the authors are studying social interactions conveyed by verbal narratives, which I would not refer to as "social interaction perception" *per se*. This made the overall framing of the paper very confusing for me. It also led to confusing arguments along the lines of "perhaps social interaction perception involves more than just perception," when a more straightforward conclusion would be that the authors are, by design, studying aspects of social interaction processing that are not perceptual.
3. Similar to point 3, the authors argue that finding similar neural effects of social interactions for written and spoken narrative presentation argues for "multimodal" processing of social interactions. Given that the social interaction information studied here is not directly present in the perceptual stimulus, but is instead inferred from language, the term "multimodal" or "modality-independent" does not seem appropriate here. A simpler explanation for these results would be that linguistic meaning is processed in a modality-independent fashion (as is well established), and that the social interaction understanding at play in this paradigm occurs downstream of language processing.

Reviewer #3

(Remarks to the Author)

The manuscript by Miao et al. investigates the neural correlates of social interaction perception and ToM using a combination of behavioral ratings and fMRI data. The study aims to determine the extent to which social interaction perception is a perceptual process or involves conceptual processes. The authors collected data from two participant groups: an online participant group provided moment-by-moment ratings of social interactions and ToM engagement while experiencing narratives, and another group underwent fMRI scanning while passively experiencing the same narratives. The ratings for social interactions and ToM engagement were reliable but only modestly correlated, suggesting that participants can distinguish between these two processes during narrative experiences. Both social interaction perception and ToM engaged overlapping brain regions, including the TPJ, STS, and mPFC. However, ToM uniquely engaged additional brain regions such as the LOTC, left aIPS, and right PMC, which the authors associated with action understanding and executive functions. The authors further discuss that social interaction perception involves both perceptual and conceptual processing, while ToM involves both pre-reflective inferences and controlled, deliberative inferences.

The presented study was generally planned and conducted carefully, and the multifaceted nature of the analyses are commendable. However, I have several questions and comments regarding clarification of methods and the interpretability of the results. In particular my questions and suggestions regarding the interpretation of results (and control for alternative explanations) aim at clarifying the degree of novelty and impact that the present study provides to the field. I think adding additional analyses and ruling out alternative explanations would greatly strengthen the study.

Methods and results:

Researcher annotations of movies:

In the methods section it is explained: "Because researchers did not experience the narratives continuously as participants did, only "yes no" (1 0) labels were used for annotations, instead of continuous values." (p. 690, 691). Can the authors please explain this more specifically. First, did the researchers rate every word in each sentence with yes/no, or each sentence of a story? What is a "sentence part" in this respect? Second, how was "tom demand" defined for researchers? In the article, the authors argue that "tom demands" should reflect something closer to stimulus features than the ratings of online participants. How was the researcher rating then defined, specifically? Also note that at this point, it is may be useful to refer to scientific definitions of tom, or explain the authors considerations. (see e.g. Quesque et al., 2024, or e.g. Schaafsma et al., 2015 for a more challenging view).

Quesque, F., Apperly, I., Baillargeon, R., Baron-Cohen, S., Becchio, C., Bekkering, H., ... & Brass, M. (2024). Defining key concepts for mental state attribution. *Communications psychology*, 2(1), 29.

Schaafsma, S. M., Pfaff, D. W., Spunt, R. P., & Adolphs, R. (2015). Deconstructing and reconstructing theory of mind. *Trends in cognitive sciences*, 19(2), 65-72.

In this respect, I also do not understand the argument that researchers only rated yes/no "because researchers did not experience the narratives continuously as participants did". How did researchers process or read the narratives? Where they segmented somehow?

Bayes Factor analysis:

In the explanation of the formula, I could not find an explanation of what the variable g reflects. It appears the formula is integrating over a distribution of g . Is g a scaling parameter or distribution?

Discussion:

Section 1:

The authors characterize the processes engaged by story reading/listening as "social interaction perception". As they also discuss, the task-format used in the present study differs from other studies presenting social interactions in the form of videos. I have the impression that the interpretation of the study could benefit from a more elaborate definition of what the authors understand as "social interaction". Did the authors also have researchers rate the videos for "social interactions"? For example are acts of communication considered as a social interaction in the same way as actions directed at another person (e.g. person A grabbing person B by the arms, as in narrative #7 in the supplementary materials)? If yes, then it would be interesting to know which of these types of social interaction are most common in the stories?

Section 3:

The authors note that several areas such as right aIPS, left PMC, bilateral LOTC or MT+ were associated with tom ratings. These areas are linked to "mirror" processes and action understanding.

I wonder if the authors can rule out alternative interpretations? For example, could low level stimulus features explain differences in brain activation? Did the authors, for example, check if sections of the narratives tagged as social-interaction-relevant by participants differ from sections of the narratives tagged as tom-relevant in low level stimulus characteristics, such as linguistic or semantic features? Linguistic features such as reading ease, word frequency, etc could be checked, as well as more "natural language processing" type of analyses, such as sentiment analyses, word semantic categorizations, or sentence complexity analyses. Many of these types of analyses could be done without testing new participants, and if the authors could show that there are no differences between story segments most highly rated for social-interaction-relevance vs. tom-relevance, also no new analyses on the neural level would need to be done. I think these kind of control analyses would significantly improve the interpretability of the results, and in particular the interpretation of the additional network the authors find for tom compared to social interactions.

Moreover, additional linguistic analyses or annotations could also allow the authors to characterize the features associated with participants ratings of "social interaction". For example, does that imply that action verbs were contained in the stories? Or that multiple persons were featured in a sentence? Is that the same for tom-ratings? Just to give one example, a classic study (Deen & McCarthy, 2010) found increased activation in the LOTC when participants read stories featuring human movement, e.g. "running down the stairs", compared to other stories.

Deen, B., and McCarthy, G. (2010). Reading about the actions of others: biological motion imagery and action congruency influence brain activity. *Neuropsychologia* 48, 1607–1615.

Version 1:

Reviewer comments:

Reviewer #1

(Remarks to the Author)

The authors have addressed the points I raised thoroughly and satisfactorily. They have also worked hard to address the points raised by the other reviewers. As far as I am concerned I consider this revised version acceptable for publication.

I have one suggestion. It would be very helpful to me and, I expect, to other readers if the authors could give examples in the main text of sentences which were: 1) social interaction only, 2) ToM only, and 3) both.

Reviewer #2

(Remarks to the Author)

The authors have addressed my concerns.

Reviewer #3

(Remarks to the Author)

All of my comments and questions have been appropriately addressed in this revised version of the manuscript. In particular, the following points have greatly improved the manuscript:

First, linguistic control analyses have been carried out for the stimulus material, and the results from these analyses show

that alternative linguistic interpretations of the results (which I pointed out as a potential concern) are highly unlikely.

Second, the authors carried out additional annotations of the stimuli (in response to my and Reviewer #2's concerns), which further improve the interpretability and specificity of their results. The neural correlates of these additional annotations reveal the nuance and complexity of the social stimulus material presented in this study, providing readers a deeper look into the neural representation of rich social interactions, going beyond label-based categorizations in terms of "theory of mind" and "social interactions".

Altogether, the authors made great efforts in revising and improving the manuscript, and in my judgment the manuscript is now in an acceptable form.

Response to Reviewers

We thank the Editor and Reviewers for your enthusiasm, thoughtful critiques, and
constructive comments. Your feedback has substantially improved the manuscript. Below, we
present an executive summary of the major changes, followed by the Reviewers' comments and
our point-by-point responses.

- • We collected new data (N = ~300) on multiple social cognitive and perceptual features
and additionally used computational linguistics models to annotate the narratives. We
compared those features with social-interaction and ToM ratings, assessed their
independent effects on neural activity, and tested whether controlling for them altered the
effects of social interactions or ToM. None of the features we tested acted as confounders
or had larger effect sizes than social interactions or ToM in predefined regions (Figures
S6-S12, Tables S1-S4). Thus, our main results are robust to these potential confounds.
- • We clarified our stance on conceptual and perceptual processing of social interactions in
verbal narratives. Because recognizing interactions in narratives is mediated by semantic
processing rather than the perceptual processing engaged by audiovisual stimuli, we
replaced “social interaction perception” with “social interaction processing” throughout. We
also related our findings to prior work on audiovisual social interactions to identify brain
regions common to perceptual and conceptual processing of social interactions.
- • We incorporated evidence on automatic components of ToM into our manuscript. We
explained how the concept of automatic ToM motivated examining ToM in naturalistic
narratives and discussed how automatic ToM may recruit regions implicated in social
interaction processing and ToM in our data.
- • To find a better balance between situating our results and avoiding unwarranted reverse
inferences, we compared our findings with established functional networks and
Neurosynth meta-analytic maps. We added these analyses to the relevant sections and
revised the Discussion accordingly.
- • We partitioned social interactions into verbal (character speaking) and nonverbal (physical)
interactions. They were similarly common in our narratives (Results) and showed
dissociable neural correlates (Figure S13). These findings do not change the main
conclusions but provide additional nuance on how social interactions are coded in the
brains of observers.
- • We made minor adjustments to statistical models that enhance statistical rigor in three
ways: (1) we replaced “standardized betas” with group-level effect sizes for associations

between behavioral features and neural activity and compared effect sizes via bootstrap
tests; (2) we mean-centered social interaction and ToM regressors within story periods,
which produced minor numerical changes from previous results, but more specific neural
correlates with all key effects that supported our conclusions; (3) we adopted a more
accurate surface-projection algorithm for better visualization. We have made
corresponding changes in Results and Discussion, including Figures 3-6 and S2-S5.

Reviewer #1

The authors have developed a most ingenious paradigm for studying social interactions.
In the first stage observers continuously rate visual or verbal presentations of social interactions,
either for perception of social interactions or for involvement of ToM. There is a very high degree
of consistency in these ratings across observers with clear distinctions between the two kinds of
rating. In the second stage volunteers are scanned while exposed to the same presentations. The
elicited brain activity can then be correlated with the ratings acquired in the first stage to indicate
brain locations showing the same fluctuations. The locations so identified are highly consistent
with previous work on social interaction perception and ToM. This paradigm has a great future. I
shall now consider some of the highlights and problems of the study.

1-1. Comparison of visual and verbal presentations. The activity elicited by these two kinds of
presentation was very similar, suggestion that the mode of presentation is not too critical.
Interestingly, verbal presentations elicited activity in early visual areas. Was this a result of visual
imagery? Do listeners report this? Another interesting observation was that similarity of verbal
and visual presentations was reduced when talking into account ToM. This is consistent with the
idea that ToM is a higher cognitive level of more abstract processing.

Thank you for commenting on this. Yes, social interaction content in verbal stimuli was
associated with neural activity in the early visual cortex (V3 and V4) and inferotemporal cortex
(Figure 3a), even when there was no visual stimulus on the screen during listening (in the Audio
modality; Page 11). In addition, levels of ToM engagement were associated with neural activity in
V3, V4, and a small part of the inferotemporal cortex (V8), both when modeled alone (Figure 4a
and Table S7) and when we controlled for the effects of social interactions (Figure 5b and Table
S10). These regions have been implicated in visual imagery. For example, neural patterns in V4
responding to color stimuli can decode the color of an imagined object (Bannert & Bartels, 2018),
and neural activity in the extrastriate cortex can decode the identity of objects during imagery (Lee

et al., 2012). It is thus possible that the neural correlates in early visual areas in the current study
were driven by visual imagery.

More generally, our findings indicate that portions of the early visual cortex and
inferotemporal cortex are sensitive not only to visual stimuli, but also to higher-order, amodal,
conceptual information. It is consistent with findings that neural patterns in primary visual cortex
can distinguish between nonvisual stimuli, such as pain versus touch (Liang et al., 2013) and
emotion categories in pictures and videos (Kragel et al., 2019). The amodal neural
representations in early visual cortices may reflect internal integrations and reorganizations of
external inputs instantiated by top-down neural circuitry, not just simple projections of external
inputs.

Unfortunately, we did not collect direct participant-level reports of visual imagery. Future
studies could manipulate visual imagery during social cognition. For example, a priming phase
before the experiment could ask participants to imagine visual scenes, versus think about possible
paraphrasing, when reading short social interaction stories; the contrast between the two
conditions could help locate neural correlates specific to visual imagery controlling for social
interaction processing. Alternatively, noninvasive brain stimulation methods (e.g., tDCS and TMS)
can be used to alternate the activity levels in brain regions causally related to visual imagery, such
as the left lateral temporal cortex (Moro et al., 2008); subsequent changes in the neural correlates
of social interaction processing and ToM could be due to changes in visual imagery processes.

We have added a brief discussion of some of these points to the manuscript on Page 25
(highlighted in blue and reproduced below).

“Intriguingly, the common neural correlates of social interaction processing and ToM also
included parts of the early visual cortex (V3 and V4) and ITC. This finding suggests that the
ventral visual stream is involved in processing amodal, conceptual information in social
narratives. The specific information represented there could be visual or mental imagery of
story content (Bannert & Bartels, 2018; Lee et al., 2012) or story-elicited emotional concepts
(Kragel et al., 2019). Future studies are needed to investigate what information causally
activates those regions during narrative processing.”

1-2. The authors make an interesting and important distinction between engagement and demand
of ToM. They show that observers can make this distinction. However, I don't think this design
allows brain areas associated with demand to be revealed. A design might be used in which there
was a period during which volunteers anticipated having to do a ToM task, but without the actual
task. This should elicit activity associated with demand.

We are thankful for the opportunity to clarify how we define and measure ToM demands.
We use the word “demand” in a different sense from what the reviewer mentioned. Demand could
mean, as the reviewer notes, an anticipated future need to engage in a mental process. Here,
however, it denotes a moment-by-moment need to use ToM to understand characters in a
story. This sense of “demand” is consistent with how several groups have defined moments in
which ToM is likely to be elicited by naturalistic stimuli. For example, Jacoby and colleagues (2016)
defined “mentalizing” events as those in which viewers were led to think about the mental states
of the characters (also see Richardson et al., 2018; Mangnus et al., 2024); Lee Masson and Isik
(2021) defined ToM events as those in which characters in the movies were thinking about the
mental states of other characters (characters were using their ToM). Both annotations share the
rationale of identifying moments in stimuli that provide “affordances” for mental state inferences
and that give participants “needs” or “demands” to use their ToM. We followed this rationale and
annotated needs/demands for ToM as the moments in the narratives with explicit mentions of
characters’ mental states. This definition is more objective than determining when participants are
“led” to use ToM (Jacoby et al., 2016) and broader than needs for “second-level ToM” (Lee
Masson & Isik, 2021), while maintaining the core definition of ToM (Quesque et al., 2024). This
defines “ToM demands” in our manuscript.

To make our definition clearer, we have added relevant content to the sections where we
introduce and define ToM demands on Pages 6, 13, and 29 (highlighted in blue).

1-3. There is a brief mention of action understanding and the mirror system. To what extent does
action understanding overlap with social interaction perception. The authors might say a bit more
about this.

Thank you for this excellent question. In our narratives, the descriptions of actions
(including intentional actions, all actions, and biological motion) do not covary with participants’
ratings of social interactions (see Response 2-1 and Supplementary Materials for details). Also,
action verbs do not predict social interaction ratings (see Response 3-4 and Supplementary
Materials). These results indicate that recognition of actions and recognition of social interactions
are distinct.

At the neural level, our new analyses showed that the effects of processing actions on
neural activity partially overlapped with those of processing social interactions but also showed
opposite effects in some regions (Figures 3 and S6). Moreover, we compared the neural
correlates of social interaction processing found here to the Neurosynth-based meta-analytic
association map of the term “action observation”. We found that the Pearson correlations between

the two maps were small (spatial $r_s = .06, .06$, in the Audio and Text modalities, respectively).
The action observation map was significant in regions such as lateral occipitotemporal cortex and
IPS, where social interaction effects were not significant. Together, the effects of social interaction
processing are distinct from action understanding.

We have added related contents to the Main Text on Page 19 (highlighted in blue and
reproduced below) and to the Supplementary Materials on Pages 9-10, 20, 22, and 25-26.

“For example, neither ratings of actions nor action verbs correlated with social interaction
ratings, and their neural correlates showed different patterns from those of social
interactions (Figure S6), suggesting a separation between social interaction processing and
action understanding.”

1-4. I was concerned that there was quite a bit of reverse inferencing. For example, in relation the
extra activity associated with ToM. Could authors justify this better.

We appreciate this comment and we are sensitive to issues of reverse inference. For
example, our group has devoted several chapters of our fMRI methods book
(<https://leanpub.com/principlesoffmri>) to discussing these issues. We have tried to strike a
balance, drawing meaningful links between our findings and other studies to interpret them in
context, while avoiding direct psychological inferences from brain findings. To address concerns
about reverse inference in our manuscript, we have gone through and edited it to describe brain
regions anatomically whenever possible, along with references to resting-state functional
networks (Yeo et al., 2011). We also removed references to reverse inferences that might be
problematic. See changes on Pages 11, 15, 18, 20-21, 24, and 26 (highlighted in blue).

To supplement this effort, we have added an analysis using Neurosynth-based term maps
and topic maps associated with psychological domains to help characterize the results. See
changes on Pages 22, 24, and 26 (highlighted in blue). We do not believe that this licenses
psychological inference from brain maps alone, but it can be a helpful guide for interpreting the
patterns of associations between our findings and those of other studies.

1-5. The authors suggest that ToM is a system 2 process that is slow and effortful. While social
interaction perception is a system 1 process. However, there is a literature, starting with Apperly
& Butterfill Psych Rev 2009, suggesting that there is system 1 type version of ToM (see Schneider
et al. Cognition, 2017). Do the authors see this process as part of social interaction perception?
Or do they need to allow for an automatic version of ToM. I note that in line 556 the authors

suggest that ToM ‘may involve more automatic processing than was previously thought’. Yet,
clearly others were previously thinking this.

We are thankful for this excellent suggestion. There are indeed findings and theories that
support an automatic component or version of ToM, such as a cognitively efficient system that
supports reasoning about belief-like states (Apperly & Butterfill, 2009). The existence of an
automatic or implicit ToM can explain the ability of human infants and some nonhuman primates
to pass ToM tasks and has been revealed in human adults in their unintentional, unconscious
processing of others’ false beliefs and visual perspectives (Schneider et al., 2017). Although
evidence for automatic ToM is still accumulating (Rakoczy, 2022), it has provided a useful
perspective for improving neuroimaging research on ToM. For example, neuroimaging research
has predominantly used controlled tasks (e.g., the false belief task; Saxe & Kanwisher, 2003) to
induce ToM, which implicitly treats ToM as a single-system, controlled process and prevents us
from investigating the neural correlates of potential automatic ToM. The theory that an automatic
ToM system may support fast, online inferences of beliefs has motivated studies that investigate
the neural correlates of “ToM events” in naturalistic experience (e.g., Jacoby et al., 2016;
Richardson et al., 2018; Mangnus et al., 2024). Conversely, comparing continuous, moment-by-
moment use of ToM during naturalistic experience with other mental processes and their neural
correlates can provide hints about the neural correlates of automatic ToM. For example, our
results showed that social interaction processing activated most ToM regions and suggested
shared pre-reflective inferences between social interaction processing and ToM. Under the
assumption that these inferences are also unconscious and support understanding of mental
states, we could infer that they are a key indicator of automatic ToM. However, our paradigm did
not allow a clean separation between automatic and controlled processing and thus could not
provide a definitive conclusion about automatic ToM. We have added discussion to elaborate on
these points on Pages 5 and 25 (highlighted in blue).

Reviewer #2

Miao et al. present an fMRI study investigating responses to social interactions and theory
of mind (ToM) during narrative comprehension. They use audio and text-based stories combined
with moment-to-moment behavioral ratings of social interaction and ToM from behavioral
participants. In a separate group of participants, they measure fMRI responses to the same
narratives, and evaluate how neural responses relate to interaction and ToM content. They find
responses to both social interaction and ToM content in a similar set of brain regions, which have

been previously implicated in social cognition. Overall, I found the study interesting and
methodologically sound. I have some reservations about the strength of the conclusions given
the possibility of confounding factors driving responses to features in naturalistic stimuli, as
described in the points below.

MAJOR POINTS

2-1. The core claims made by the paper rely on relationships between fMRI responses and
abstract features (social interactions, ToM) in complex, naturalistic story stimuli. Given the
complex nature of the stimulus, there are many potential confounding stimulus features that could
be related to the features of interest: for instance, the presence of people at all, the number of
people in a scene, the presence of body movements or intentional action, etc. These various
confounding factors are not measured or modeled in the analysis. It is therefore not clear that the
effects ascribed to social interaction and ToM in this paper don't result from an influence of
confounding factors. The authors could make a stronger case for effects specific to social
interaction if they were to quantify multiple features relevant to social cognition, and assess their
independent effects on fMRI responses.

We are thankful for the reviewer's comments and the opportunity to further strengthen our
findings and arguments. We agree that stimulus features related to social interactions and/or ToM
ratings could confound their neural correlates. To address this, we collected additional data, ran
further analyses on multiple features, and tested whether any of these features are plausible
confounds. The social cognitive features we tested included intentional actions, all actions,
biological motion (or body movements), the presence of multiple people in a scene ("multi-person
presence"; about 100 participants rated each of these features), indoor versus outdoor scenes
(manually annotated by the authors), and the valence of narratives (quantified by sentiment
polarity predicted by a large language model). We also evaluated various potentially related
linguistic features extracted with models from computational linguistics (please see Reviewer
Comment and Response 3-4 for details).

We assessed whether each feature was a potential confounder that could cause spurious
relationships between social interactions/ToM and neural activity, considering both theoretical
and statistical evidence (Cinelli et al., 2024; Pearl, 2021). Theoretically, because we are
investigating the total effects of social interactions/ToM on brain activity, a confounder must
causally precede and influence social interactions/ToM and the fMRI signal. Downstream
consequences of social interactions/ToM are not confounders; controlling for them would bias

estimates of the total effect (Cinelli et al., 2024). Variables unrelated to social interactions/ToM
cannot influence them and thus cannot be confounders (Cinelli et al., 2024). Accordingly, we first
tested whether each feature was statistically correlated with social interactions and ToM and had
overlapping neural correlates with them. For features that were related and plausible antecedents,
we assessed the effects of social interactions/ToM on fMRI data while controlling for each feature.
In addition, we compared the absolute effect sizes of each variable's relationship with neural
activity to those of social interactions/ToM in predefined brain masks associated with social
interactions or ToM in prior meta-analyses. This tested whether any potential confounders were
better predictors of brain activity than social interactions and/or ToM.

We now report the new online studies used to collect data, new analyses, and results on
Pages 2-5 and 8-29 of the Supplementary Materials, as well as a summary on Pages 18-19 of
the Main Text (highlighted in yellow). To summarize (see Tables S1-S2, S4 and Figures S6-S12),
we found that intentional actions, all actions, biological motion, the valence of narratives (Table
S2, reproduced below), and linguistic features (Table S4, reproduced below) were not reliably
correlated with social interactions or ToM ratings. Indoor versus outdoor scenes were associated
with social interactions and ToM ratings only in the Text modality and showed some overlapping
neural correlates, but statistically controlling for them had little effect on the activity maps of social
interactions and ToM. Multi-person presence correlated strongly with social interactions and
moderately with ToM, and its effect maps were very similar to those of social interactions and
ToM. However, social interactions still explained neural activity in almost all key brain regions
after controlling for multi-person presence, and the effects of ToM were only slightly affected by
controlling for multi-person presence. In addition, none of the effects of the social cognitive
features on fMRI responses in predefined brain masks were significantly larger than those of
social interactions or ToM, and most were significantly smaller (Figure S12 and Table S1).
Therefore, our findings about social interactions and ToM are robust to potential confounding
features. In addition to these results, we will make the full set of annotations available to the
research community as a supplementary file upon publication, which can be paired with the open
fMRI dataset (Jung et al., 2025) and enable replications or further analyses. In the following
sections, we provide details of the methods and results associated with each social cognitive
feature (we have added those contents to Supplementary Materials; for linguistic features, see
Response 3-4).

**Table S2**

*Pearson’s Correlations between social cognitive features and ratings of social*
 *interactions and ToM across time*

Features	Social interactions			ToM		
	Audio	Text	All	Audio	Text	All
Intentional actions	-.02	.11	.06	.03	.03	.03
All actions	-.10	.14	.06	.02	-.03	-.01
Biological motion	-.05	.09	.04	.11	-.07	-.003
Multi-person presence	.70	.81	.77	.26	.29	.28
Indoor – outdoor	.03	.20	.14	-.05	.27	.16
“Positive” label	.01	-.08	-.04	-.01	.01	-.001
“Neutral” label	.17	-.09	-.004	.01	-.19	-.12
“Negative” label	-.23	.12	.03	-.002	.19	.13
“Positive” probabilities	.03	-.07	-.05	.03	-.03	.004
“Neutral” prob.	.10	-.08	.01	-.05	-.13	-.08
“Negative” prob.	-.14	.11	.02	.03	.14	.08

*Note.* “Text” and “Audio” indicate correlations within Text and Audio narratives, respectively, and
 “All” indicates correlations across all narratives.

**Table S4**

*All linguistic features annotated, the methods of annotations, and their Pearson’s*
 *correlations with ratings of social interactions and ToM*

Linguistic features	Annotation method (package used)	Social interactions			ToM		
		Audio	Text	All	Audio	Text	All
Reading ease – Word frequency	“wordfreq” in Python (Robyn Speer, 2022)	-.02	-.01	-.02	-.03	-.06	-.06
Reading ease – Number of syllables	“textstat” in Python ¹	.03	.07	.06	.01	.04	.03
Part-of-speech (POS)	“flair” in Python (Akbik et al., 2018)	<.05	<.04	<.01	<.03	<.06	<.04
		.01 (all POS prediction)			.06 (all POS prediction)		
Sentence complexity	“textstat” in Python	.09	.03	.05	-.03	-.0003	.01
Semantic frames – “Statement”	“frame-semantic-transformer” in Python (Chanin, 2023)	.18	.39	.32	.16	.13	.14

¹ <https://pypi.org/project/textstat/>

Semantic frames – Other frames		<.12	<.09	<.09	<.11	<.13	<.12
--	------	------	------	------	------	------

275 *Note.* “Text” and “Audio” indicate correlations within Text and Audio narratives, respectively, and
276 “All” indicates correlations across all narratives. Cells with a “<” indicate that the correlations
were calculated between social interactions/ToM and more than one features (five POS tags
and seven frames) and the absolute values of all correlations were below the value shown
there.

Intentional actions and all actions

Following previous studies (Pelphrey et al., 2004; Iacoboni et al., 2005; Desmet & Brass,
2015; Kana et al., 2015), we defined intentional actions as those that involve clear motivation, a
certain level of planning, and congruence with the context. We defined accidental actions as
actions that are not planned or willed by the actors, providing a counterpoint to intentional actions
(Kana et al., 2015). We also defined all actions as the union of intentional and accidental actions.
Because the presence of intentional actions can be ambiguous in language (Mele & Cushman,
2007), we recruited a group of participants ($N = 101$, 52 females, mean age = 34.38 years ($SD =$
6.62)) to judge whether each small part of the narratives contained “intentional actions,”
“accidental actions,” or “no actions” (for details, please see Supplementary Methods – Behavioral
ratings of action-related features). We then quantified intentional actions in narratives as the
proportion of participants who provided a judgment of intentional actions, while all actions were
the sum of the proportions of intentional and accidental action judgments.

We found that levels of intentional actions correlated very weakly with social interactions
(for the Text modality, $r = .11$; for the Audio modality, $r = -.02$; across all narratives, $r = .06$) and
with ToM ($r_s = .03, .03, -.03$), as well as with all actions (all absolute $r_s < .11$). We tested the
independent effects of processing intentional actions and all actions on fMRI responses. The
neural correlates of processing intentional actions were similar across the Audio and Text
modalities (the spatial correlation between the two unthresholded t maps or “spatial r ” = .64), and
less so for all actions (spatial $r = .45$). The neural correlates of processing intentional actions and
all actions were very similar (spatial $r_s = .94$ and $.76$ for the Audio and Text modalities,
respectively). Across modalities, both were positively associated with neural activity in the bilateral
medial temporal lobes (MTL), posterior cingulate cortex (PCC), bilateral supramarginal gyrus
(SMG), left angular gyrus (AG), and right inferior frontal sulcus (IFS), and negatively associated
with the left TPJ, left dmPFC, bilateral anterior STS, left middle STS, and left IFG (Figure S6a-b).

Biological motion

We defined biological motion as actions that have explicit descriptions of the movement
of human bodies or body parts (e.g., “run”, “attack”, “fall”; see Deen & McCarthy, 2010). Using a
paradigm similar to that for intentional action judgments, we collected judgments about biological
motion from another group of participants ($N = 106$, 51 females, mean age = 33.27 years ($SD =$
6.64)) and quantified the levels (probabilities) of biological motion as the proportions of
participants who gave a judgment of biological motion.

We found that levels of biological motion correlated very weakly with social interactions
(for the Text modality, $r = .08$; for the Audio modality, $r = -.05$; across all narratives, $r = .04$) and
ToM ($r_s = .11, -.07, -.003$). By contrast, biological motion strongly correlated with intentional
actions (all $r_s > .60$) and all actions (all $r_s > .75$), implying that a large proportion of actions in the
narratives are biological motion. At the neural level, the effects of biological motion were similar
across modalities (spatial $r = .59$). In both modalities, processing biological motion was positively
associated with neural activity in the right TPJ, right pSTS, left AG, left SMG, bilateral superior
parietal lobule, MTL, and PCC. It was negatively correlated with bilateral anterior STS, PC, left
IFG, left dmPFC, left TPJ, left middle STS, and bilateral ventromedial prefrontal cortex (Figure
S6c). The effect maps of biological motion were also similar to those of intentional actions and all
actions (all spatial $r_s > .60$). All three maps shared positive regions with social interactions and
ToM, such as the left AG, but they also contained negative clusters in regions positively related
to social interactions/ToM, such as the left TPJ and bilateral PC. These results suggest
dissociable neural correlates between social interactions/ToM and action-related features.
Together with the low correlations between ratings, they indicate that intentional actions, all
actions, and biological motion do not confound the effects of social interactions and ToM.

**Figure S6.** Neural correlates of processing Intentional actions, All actions, and Biological motion
as conjunctions across two modalities. Those neural correlates partially overlapped with those of
social interactions and ToM, but also had an opposite sign in other regions, suggesting their

effects were different from those of social interactions and ToM. The three maps were thresholded
at FDR $q < .01$ independently.

People-related features (primarily multi-person presence)

People are what make experimental stimuli “social”, so it is possible that some of the
neural correlates we observed for social interactions and ToM are caused by the presence of
people. However, because our narratives are all about human characters and their stories, **every**
**sentence** in the narratives references people, as indicated by character names, pronouns,
possessive adjectives, and so on. With more fine-grained annotations, fewer than ten clauses in
our narratives did not include people (e.g., “weeks passed”, “everything was going according to
the plan”). Thus, the feature of the presence of people is virtually constant across all our stimuli
and cannot confound our findings.

We further considered the number of people in a scene. Unlike visual stimuli that can be
cut into frames where the number of people is readily clear, our verbal narratives do not have
many natural scene boundaries, and it can be hard to determine how many people are in a scene.
Indeed, different readers can form different mental images of the same text with different numbers
of people in it. Thus, instead of trying to find the exact number of people present in the narratives,
we tested the feature of the presence of multiple (more than one) people, or multi-person
presence, which was also suggested by another reviewer in Comment 3-4.

To quantify multi-person presence, we asked another group of participants ($N = 103$, 50
females, 19-45 years old ($M = 32.02$, $SD = 6.53$)) to continuously experience the narratives and
provide ratings of the extent to which multiple people were present at each moment (for method
details, please see Supplementary methods -- Behavioral ratings of multi-person presence).
Ratings of multi-person presence were highly reliable (mean split-half correlation = .96). We used
the medians of all participants’ ratings to as the normative measure of multi-person presence. It
correlated strongly with social interactions ($r_s = .70$, .81, for the Audio and Text modality,
respectively; $r = .77$ across all narratives; Figure S7a) and weakly to moderately with ToM (r_s
= .26, .29, .28). At the neural level, the independent effects of multi-person presence were similar
across modalities (spatial $r = .64$) and similar to those of social interactions (spatial $r_s = .86$, .96
for Audio and Text modality, respectively) and ToM (spatial $r_s = .83$, .75). Common positive neural
correlates across modalities included bilateral TPJ, STS, PC, dmPFC, and IFG; common negative
neural correlates were rare (Figure S7b).

a) Ratings of multi-person presence and social interactions

b) Neural correlates of processing multi-person presence

c) Neural correlates of processing multi-person presence controlling for

**Figure S7.** Results about the presence of multiple people (multi-person presence). a) The
 medians of online participants' ratings of multi-person presence and social interactions across all
 story timepoints. The shaded areas around the lines are simulated confidence intervals calculated
 by median absolute differences. Ratings of multi-person presence and social interactions were
 highly correlated, with the former being higher or similar to the latter at almost all time points. b)
 The group-level t maps of processing multi-person presence, thresholded at FDR $q < .01$. They
 were very similar to the maps of social interactions and ToM. c) Group-level t maps of processing
 multi-person presence after controlling for social interactions or ToM, as a conjunction across two
 modalities, thresholded at FDR $q < .01$. Compared to its independent effects, controlling for social
 interactions rendered most significant voxels insignificant, while controlling for ToM had smaller
 effects.

Theoretically, multi-person presence is a necessary but not sufficient condition for social
 interactions and may be considered part of the construct of social interaction itself. This makes it

conceptually problematic as a covariate. Formally, in the causal model “Brain -> Perception of
multiple people -> Recognition of social interaction”, multi-person presence is a mediator, and
controlling for it will induce a bias by removing part of the causal effect (Cinelli et al., 2024). Multi-
person presence could also be a common effect of brain activity in relevant regions and social
interactions (e.g., a collider) if the perception of social interactions causes one to believe that
multiple people are present (e.g., imagine entering a house and hearing one person speaking;
one might infer that another person is present). If so, controlling for multi-person presence will
introduce bias in the estimate of the brain-social interaction relationship (Cinelli et al., 2024).
Indeed, although some controlled experiments have matched the number of people in interaction
and no-interaction conditions (e.g., Isik et al., 2017; Landsiedel et al., 2022), naturalistic studies
of social interaction processing have rarely done so (e.g., Lahnakoski et al., 2012; Lee Masson &
Isik, 2021). These theoretical considerations motivated our decision to present analyses
controlling for multi-person presence only with these caveats in mind. We compared the neural
correlates of social interactions with and without controlling for multi-person presence, which, as
might be expected, reduced *t* values for social interaction effects across the brain. However, 58.6%
(Audio) and 30.8% (Text) of the positive significant voxels associated with social interactions were
still significant with the same sign, using the same corrected threshold (Figure S8a). Significant
regions also covered most key brain regions of interest, including bilateral STS, dmPFC, IFG, left
PC, and left TPJ; the only exception was the loss of significance in right TPJ (Figure S8a). The
overall activity patterns also remained stable after controlling for multi-person presence (spatial
*r*s = .81 and .69 for the Audio and Text modality, respectively). By contrast, only 21.6% (Audio)
and 25.7% (Text) of positive significant voxels associated with multi-person presence were still
significant after controlling for social interactions (Figure S7c).

Next, we replicated the same analyses for ToM—which does not have the same
theoretical relationship with multi-person presence and is thus more straightforward as a
covariate—and found that 94.8% (Audio) and 83.3% of positive significant voxels were still
significant with the same sign while controlling for multi-person presence (Figure S8b). These
results indicate that, despite being a defining characteristic of social interactions, multi-person
presence could not explain away most of the effects of social interactions and had a weaker
influence on ToM. In other words, the neural effects of social interactions and ToM were not
attributable to the simple effect of multi-person presence.

**Figure S8.** Neural correlates of processing social interactions (a) and ToM (b) after controlling for
 multi-person presence, thresholded at FDR $q < .01$. In the scatter plots on the right, t values of
 each voxel without and with controlling for multi-person presence were plotted along the x and y
 axis, respectively; red dashed lines indicate the threshold for significance; dark purple dots
 indicate the voxels that were significant with the same sign regardless of whether to control multi-
 person presence, light purple dots indicate the voxels that were significant without controls but
 were not significant with controls, and gray dots indicate voxels that were not significantly
 associated with social interactions/ToM. This figure shows that the effects of social
 interactions were reduced by controlling for multi-person presence but still significant in
 most key regions, while the effects of ToM were rarely affected.

Indoor/Outdoor scenes

Following other studies using naturalistic stimuli (e.g., Chen et al., 2017; Lee Masson &
 Isik, 2021; McMahan et al., 2023), we manually annotated each sentence in our narratives as

occurring indoors, outdoors, or in an ambiguous location (e.g., “Linda and Amy had been best
 friends since kindergarten”). Overall, there were more indoor scenes (48.4% of all story time
 points) than outdoor scenes (34.0%) in the narratives we used, followed by ambiguous locations
 (17.6%). We took the difference between the indoor and outdoor annotations to obtain an estimate
 of the contrast feature “indoor versus outdoor”. This contrast feature did not correlate with ratings
 of social interactions ($r = .03$) or ToM ($r = -.05$) in the Audio modality, but did correlate nontrivially
 in the Text modality ($r_s = .20, .27$ for social interactions and ToM, respectively). At the neural level,
 the independent effects of the contrast were not reliable across modalities (spatial $r = .16$). Still,
 in both modalities indoor scenes activated bilateral PC and right TPJ more strongly than outdoor
 scenes, with additional effects in bilateral STS, dmPFC, IFG, and left TPJ in the Text modality
 (Figure S9). Those regions overlapped with the neural correlates of social interactions and ToM.

 **Figure S9.** Neural correlates of processing indoor scenes minus outdoor scenes, thresholded at
 FDR $q < .01$. The positive regions overlapped with the positive neural correlates of social
 interaction processing and ToM. After controlling for social interactions or ToM, there were rarely
 any significant voxels ($\sim 0.1\%$ of all voxels) in the conjunction map across modalities, so we did
 not show the effect maps here.

 We further investigated whether controlling for indoor/outdoor scenes affected the effects
 of social interactions and ToM (Figure S10). We found that almost all positive significant voxels
 associated with social interactions were still significant with the same sign after controlling for
 indoor versus outdoor scenes (94.1% and 91.9% for the Audio and Text modality, respectively),
 as well as for ToM (95.5% and 85.1%). The spatial patterns of social interaction effects also
 remained largely unchanged after controlling for indoor versus outdoor scenes (spatial $r_s =$
 $.97, .97$ for the Audio and Text modality, respectively), as did those of ToM ($r_s = .98, .90$). These
 results suggest that controlling for indoor versus outdoor scenes had little effect on the neural
 correlates of social interactions and ToM and thus is not a confounding variable.

**Figure S10.** Neural correlates of processing social interactions (a) and ToM (b) after controlling
 for the contrast between indoor and outdoor scenes, thresholded at FDR $q < .01$. The brain maps
 on the left show that the effects of social interaction processing and ToM were rarely affected by
 processing indoor versus outdoor scenes. In the scatter plots on the right, t values of each voxel
 without and with controlling for indoor-outdoor scenes were plotted along the x and y axis,
 respectively; all conventions follow those of Figure S8.

Valence of narratives

Valence is a major axis along which affective experiences vary, and it has been grouped
 with social cognitive features in existing studies (e.g., Lee Masson & Isik, 2021; McMahon et al.,
 2023). We annotated the valence of each sentence in our narratives using a BERTweet-based
 language model fine-tuned for sentiment analysis (Pérez et al., 2021). We found that neither the
 “positive”, “negative”, or “neutral” labels nor their probabilities were related to social interactions
 (all absolute r s $< .06$) or ToM (all absolute r s $< .14$) ratings across all narratives. At the neural

level, we tested the contrast effects between different sentiment labels (e.g., positive versus
 negative sentences). We found the result patterns were not generalizable across modalities (all
 spatial $r_s < .3$). In each modality, none of the three contrasts resembled the neural correlates of
 social interactions or ToM (all spatial $r_s < .25$ except one spatial $r = .44$). For example, the contrast
 between positive and negative sentences activated bilateral PCC, MTL, and orbitofrontal cortex
 in the Text modality, and left TPJ and ventral medial prefrontal cortex in the Audio modality.
 Although some contrast maps in one modality showed overlapping activations with social
 interactions or ToM (e.g., Neutral versus Negative sentences in the Audio modality), they were
 not generalizable across modalities. Together with the low behavioral correlations, these results
 indicate that valence in narratives does not confound the effects of social interactions or ToM.

 **Figure S11.** Neural correlates of the contrasts of processing different sentiment-label sentences.
 The three panels a to c show the effect maps of Positive versus Negative, Positive versus Neutral,
 and Neutral versus Negative, respectively, thresholded at FDR $q < .01$. Because maps are
 different across modalities, only a few voxels showed cross-modal activations, so no conjunction
 maps are displayed. Across all maps, some positive regions overlapped with those of social
 interactions/ToM, but those patterns were not consistent across modalities or contrasts.

 Independent effects of features in pre-defined brain masks

To investigate whether any of the above features explained neural activity better than
 social interactions or ToM in regions that have been found to be associated with social interactions

or ToM, we conducted mask-based analysis similar to the comparisons between ToM
engagement and ToM demands in the Main Text. Specifically, we created brain masks for “social
interaction”, “social cognition”, “pSTS”, “ToM”, and “mentalizing” based on Neurosynth meta-
analytic term maps. Within each mask, we calculated, for every participant, the mean beta
associated with each feature and divided it by the standard deviation across participants, yielding
a group-level effect size estimate (Cohen’s *d*). We then compared, within each mask and modality,
the absolute effect sizes for each feature with those for social interactions and ToM. Social
interactions and ToM had significantly larger effect sizes than nearly all other variables (except
multi-person presence) in some comparisons and were not significantly smaller in any comparison
(see Figure S12 and Table S1). These results indicate that, in canonical social interaction and
ToM regions, social interactions and ToM ratings in the current narratives explained neural activity
better than or as well as any other social cognitive feature.

X-axis labels: 1 – Social interactions, 2 – ToM
 3 – Intentional actions, 4 – All actions, 5 – Biological motion, 6 – Multi-person presence
 7 – Indoor vs. outdoor, 8 – Positive vs. Negative, 9 – Positive vs. Neutral, 10 – Neutral vs. Negative

**Figure S12.** The effect sizes of 10 features in explaining neural activity in pre-defined brain masks
 associated with social interactions or ToM. The y-axes showed the individual mean regression
 coefficients (betas) divided by the standard deviation across participants. The top edges of the
 box are the 1st and 3rd quartile of each group of data, the whiskers extend to the most extreme
 data points not considered outliers (data points that are more than 1.5 interquartile range away
 from the 1st and 3rd quartile), and outliers are plotted individually as '*' symbols. Solid lines within

boxes represent the means (Cohen's *d*), while dotted lines represent medians (in many cases
 they overlap in the figure). In most comparisons, social interactions and ToM had significantly
 larger effect sizes, while in any comparisons, none of the other features had larger effect sizes
 than social interactions/ToM.

**Table S1**

*Differences in absolute effect sizes of all pair-wise comparisons between social*
 *interactions/ToM and other social cognitive features*

	Inten- tional actions	All actions	Biolo- gical motion	Multi- person presence	Indoor versus outdoor	Positive versus Negative	Positive versus Neutral	Neutral versus Negative
"social interaction" mask – Text modality								
Soc.	0.94	0.95	1.03	-0.03	0.49	0.99	0.60	0.38
ToM	0.70	0.70	0.78	-0.28	0.25	0.74	0.35	0.13
"social interaction" mask – Audio modality								
Soc.	0.35	0.41	0.81	-0.03	0.87	0.96	0.68	0.90
ToM	0.60	0.65	1.06	0.21	1.12	1.20	0.92	1.15
"social cognition mask" – Text modality								
Soc.	0.76	0.93	0.73	-0.01	0.37	0.91	0.55	0.37
ToM	0.48	0.65	0.45	-0.29	0.08	0.63	0.27	0.09
"social cognition mask" – Audio modality								
Soc.	-0.06	0.07	0.37	0.02	0.88	0.81	0.51	0.83
ToM	0.06	0.18	0.49	0.14	1.00	0.93	0.63	0.94
"psts" mask – Text modality								
Soc.	0.73	0.54	0.39	-0.02	0.49	0.79	0.44	0.24
ToM	0.40	0.21	0.06	-0.35	0.16	0.46	0.11	-0.09
"psts" mask – Audio modality								
Soc.	-0.03	0.18	0.48	-0.12	0.55	-0.21	-0.18	0.31
ToM	0.36	0.57	0.88	0.27	0.95	0.18	0.21	0.70
"tom" mask – Text modality								
Soc.	0.74	0.91	0.69	-0.02	0.39	1.09	0.72	0.36
ToM	0.59	0.77	0.54	-0.17	0.25	0.94	0.57	0.21
"tom" mask – Audio modality								
Soc.	-0.06	-0.04	0.30	0.06	0.93	1.01	0.61	0.82
ToM	0.16	0.18	0.52	0.28	1.16	1.23	0.83	1.04

"mentalizing" mask – Text modality								
Soc.	0.72	0.81	0.54	-0.03	0.37	1.01	0.65	0.37
ToM	0.62	0.71	0.44	-0.12	0.27	0.91	0.55	0.27
"mentalizing" mask – Audio modality								
Soc.	-0.02	-0.02	0.32	-0.002	0.86	1.06	0.59	0.68
ToM	0.23	0.23	0.57	0.25	1.11	1.31	0.84	0.93

*Note.* Bold fonts indicate values that are significant under the Bonferroni Corrected (160
comparisons) threshold of .05 (p values estimated from 10,000 Bootstrap samples). Positive
values indicate that social interactions/ToM have larger effect sizes. Abbreviations: Soc. – Social
interactions. The table shows that social interactions and ToM have larger absolute effect sizes
in the vast majority of comparisons, and most of them are statistically significant; no comparisons
show significantly smaller absolute effect sizes of social interactions/ToM

2-2. The authors frame their results in terms of "social interaction perception." In the literature,
this term has typically been used to describe inferences about social interaction directly from
visual or auditory cues. In the current manuscript, the authors are studying social interactions
conveyed by verbal narratives, which I would not refer to as "social interaction perception" per se.
This made the overall framing of the paper very confusing for me. It also led to confusing
arguments along the lines of "perhaps social interaction perception involves more than just
perception," when a more straightforward conclusion would be that the authors are, by design,
studying aspects of social interaction processing that are not perceptual.

We appreciate the excellent comments and suggestions, and we have revised the
manuscript and our use of terminology to clarify. We agree that we are studying the conceptual
aspects of social interaction processing, while we also note that similar conceptual processing
happens when people perceive social interactions from visual and/or auditory stimuli, and our
findings can be linked and compared to them. In our prior manuscript, we defined "perception"
broadly, as "a way of regarding, understanding, or interpreting something; a mental impression"
(Oxford Language²). In psychology and neuroscience literature, many theories and empirical
findings point to the important role of conceptual processing in almost all types of perception. For
example, people often cannot identify objects in two-tone "Mooney images" (Mooney, 1957) until
they know what the object is, which can dramatically facilitate recognition (Gregory, 1970). Once
objects in such figures are disambiguated, they are hard to "unsee" (Ludmer et al., 2011). This
influence of prior knowledge on perception occurs in a fraction of a second (Flounders et al., 2019;

² <https://languages.oup.com/google-dictionary-en/>

Hardstone et al., 2021) and can last for many days (Ludmer et al., 2011). Conceptual information
can lead to rapid and long-lasting influences on perception in other modalities as well. Such
findings dovetail with theories of predictive coding in perception (Rao & Ballard, 1999; Clark,
2013), in which percepts are produced by comparing representations stored in memory (the
internal model) with incoming sensory inputs. Thus, to recognize social interactions from
movements of geometric shapes (e.g., Heider & Simmel, 1944; Varrier & Finn, 2022), light points
(e.g., Isik et al., 2017), human figures (e.g., Landsiedel et al., 2022), and other sensory inputs,
people need to infer higher-level motion statistics and match them to stored conceptual patterns
(McMahon & Isik, 2023). The conceptual processes required to identify objects as well as more
abstract phenomena like social interactions are tightly integrated with perceptual processing. We
reason that linguistic descriptions of social interactions can directly evoke their associated
conceptual patterns in the brain without needing to compute higher-order statistics from sensory
inputs. They can further engage the conceptual processing in social interaction perception.

Despite that, it is true that the use of “social interaction perception” in the current study
can be misleading and confused with existing literature that uses visual and/or auditory stimuli.
To avoid confusion, we now refer to this process as “social interaction processing” (Sliwa &
Freiwald, 2017; Arioli & Canessa, 2019) and define it as the emergence of the perception of social
interactions from the combined effect of sensory and conceptual inputs. It emphasizes the
contributions of both perceptual and conceptual experience, without being tied to specific sensory
modalities or restricted to only perceptual processes. We also reframed our arguments about
social interactions: instead of arguing that social interaction perception involves conceptual
processing, we now ask whether the conceptual aspect of social interaction processing engages
brain regions shared with those found by audiovisual presentations of social interactions. We
have made changes accordingly throughout the manuscript, especially in the Title, Introduction,
and Discussion (key changes highlighted in yellow).

2-3. Similar to point 3, the authors argue that finding similar neural effects of social interactions
for written and spoken narrative presentation argues for "multimodal" processing of social
interactions. Given that the social interaction information studied here is not directly present in the
perceptual stimulus, but is instead inferred from language, the term "multimodal" or "modality-
independent" does not seem appropriate here. A simpler explanation for these results would be
that linguistic meaning is processed in a modality-independent fashion (as is well established),
and that the social interaction understanding at play in this paradigm occurs downstream of
language processing.

Thank you for the comment. We agree that linguistic meaning is processed in a modality-
independent fashion, which can account for the correlation between the effect maps of social
interactions in the Text and Audio modality to some degrees. At the same time, our neural results
showed that social interactions in narratives activated both higher-level multimodal regions and
lower-level visual regions, even in the absence of visual stimuli. Importantly, these regions
substantially overlap with previous findings on audiovisual social interactions, such as posterior
STS (e.g., Isik et al., 2017) and dmPFC (Wagner et al., 2016), even for simple visual stimuli such
as geometric shape animations (Heider & Simmel, 1944; Varrier & Finn, 2022). These findings
provide evidence that these overlapping regions process social interactions regardless of
presentation modality. They further point to heteromodal processing in perceiving social
interactions, congruent with previous findings (Landsiedel & Koldewyn, 2023; Lee Masson et al.,
2024) and our contention that verbal descriptions of social interactions induce shared conceptual
processing involved in perceiving audiovisual social interactions.

To avoid confusion, we now treat the correlation of social interaction effect maps across
modalities as a feature of our task, instead of a piece of evidence for the modality-independent
processing of social interactions. For example, we have changed one sentence in Results into
the following:

“This suggests that participants use overlapped neural circuitries to process social
interactions both when reading and when listening to the narratives, consistent with the
modality-independent nature of conceptual processing.”

Other related changes in Results and Discussion are highlighted in yellow on Pages 7, 11,
and 24.

Reviewer #3

The manuscript by Miao et al. investigates the neural correlates of social interaction
perception and ToM using a combination of behavioral ratings and fMRI data. The study aims to
determine the extent to which social interaction perception is a perceptual process or involves
conceptual processes. The authors collected data from two participant groups: an online
participant group provided moment-by-moment ratings of social interactions and ToM
engagement while experiencing narratives, and another group underwent fMRI scanning while
passively experiencing the same narratives. The ratings for social interactions and ToM
engagement were reliable but only modestly correlated, suggesting that participants can
distinguish between these two processes during narrative experiences. Both social interaction

perception and ToM engaged overlapping brain regions, including the TPJ, STS, and mPFC.
However, ToM uniquely engaged additional brain regions such as the LOTC, left aIPS, and right
PMC, which the authors associated with action understanding and executive functions. The
authors further discuss that social interaction perception involves both perceptual and conceptual
processing, while ToM involves both pre-reflective inferences and controlled, deliberative
inferences.

The presented study was generally planned and conducted carefully, and the multifaceted
nature of the analyses are commendable. However, I have a several questions and comments
regarding clarification of methods and the interpretability of the results. In particular my questions
and suggestions regarding the interpretation of results (and control for alternative explanations)
aim at clarifying the degree of novelty and impact that the present study provides to the field. I
think adding additional analyses and ruling out alternative explanations would greatly strengthen
the study.

Methods and results:

3-1. Researcher annotations of movies:

In the methods section it is explained: "Because researchers did not experience the narratives
continuously as participants did, only "yes no" (1 0) labels were used for annotations, instead of
continuous values." (p. 690, 691). Can the authors please explain this more specifically. First, did
the researchers rate every word in each sentence with yes/no, or each sentence of a story? What
is a "sentence part" in this respect?

We appreciate these clarifying questions. Researchers' annotations were done in units of
"sentence parts", the smallest chunk in a sentence that contains one predicate or a sequence of
predicates and their associated objects (Pages 14 and 29-30 of the Main Text). This definition
and analysis choice are motivated by prior literature on annotating features in verbal narratives,
such as Nijhof & Willems (2015). For example, the sentence "Shortly after, Margaret heard about
Al's confession and infidelity on the local news" has only one sentence part because there is only
one predicate, "heard"; by contrast, the sentence "She was furious that she did not know what
was going on and hurt that Al had been acting so selfishly behind her back" has two sentence
parts because there are two predicates, "was furious" and "hurt", and their associated objects.
Each word in the narratives was assigned to one sentence part (for a full list of sentence parts,
see Supplementary Materials).

For annotating the narratives, four researchers in our group (including the first author)
independently viewed each sentence part in the same order as in narratives and determined
whether it contained ToM demands (yes/no or 1/0). A "consensus label" for each sentence part

was generated based on those annotations, and conflicts (where only two annotators agreed)
were resolved by further discussion. Thus, each word in a sentence part was assigned to the
same label. We have revised the manuscript where the concept of sentence parts and researcher
annotations is first introduced (Page 14, reproduced below) and added more details about these
annotations in Methods (Pages 29-30; all changes highlighted in green).

“To obtain ratings for ToM demands, four researchers in our group independently annotated
the narratives. We first cut the narratives into sentence parts, the smallest chunk in a
sentence that contains one predicate or a sequence of predicates and their associated
objects (Nijhof & Willems, 2015; see Supplementary Materials for a full list of sentence parts).
Then the raters read the sentence parts in their original order and determined whether each
sentence part had explicitly mentioned mental states of characters (1 = yes, 0 = no), as the
labels for “ToM demands” (see Figure 1b for an example).”

Second, how was "tom demand" defined for researchers? In the article, the authors argue
that "tom demands" should reflect something closer to stimulus features than the ratings of online
participants. How was the researcher rating then defined, specifically? Also note that at this point,
it is may be useful to refer to scientific definitions of tom, or explain the authors considerations.
(see e.g. Quesque et al., 2024, or e.g. Schaafsma et al., 2015 for a more challenging view).
Quesque, F., Apperly, I., Baillargeon, R., Baron-Cohen, S., Becchio, C., Bekkering, H., ... & Brass,
664 M. (2024). Defining key concepts for mental state attribution. *Communications psychology*, 2(1),
29.

Schaafsma, S. M., Pfaff, D. W., Spunt, R. P., & Adolphs, R. (2015). Deconstructing and
reconstructing theory of mind. *Trends in cognitive sciences*, 19(2), 65-72.

In this respect, I also do not understand the argument that researchers only rated yes/no
"because researchers did not experience the narratives continuously as participants did". How
did researchers process or read the narratives? Where they segmented somehow?

We defined ToM demands as “whether each sentence part had explicitly mentioned
mental states of characters”, where mental states “included both the cognitive aspects (e.g.,
beliefs and goals), and the affective aspects (e.g., emotions and desires)” (Page 13). This
definition served as the criterion to annotate ToM demands. It reflects the moment-by-moment
need for people to use ToM to understand the characters in our narratives (see also Response
1-2). Although the precise definition and measurement of ToM vary across studies (Schaafsma
et al., 2015) and there is still a need for consensus among researchers (Quesque et al., 2024),
understanding the mental states of other people is shared across all definitions. In naturalistic

experience, it is hard to determine which specific cognitive processes people use to support
mental state inference (e.g., whether to inverse a conceptual model of mental states; Quesque et
al., 2024), but the needs or affordances brought about by mental state descriptions are much
clearer and easier to define (Jacoby et al., 2016; Richardson et al., 2018; Mangnus et al., 2024).
Thus, we used the presence of mental states as a generic and objective definition of ToM
demands. We have clarified the definition and its motivation where “ToM demands” is first
introduced (Pages 6 and 13, related to Response 1-2 and highlighted in blue). We have also
added discussions about how ToM demands relate to different definitions of ToM on Page 25
(highlighted in green) and reproduced below:

“Third, to allow more objective annotations for ToM demands and easier self-report of ToM
engagement, we adopted a more generic definition of ToM as the mental process of thinking
about other people’s mental states. Although it captures the core definition of ToM, it cannot
reveal different sub-processes involved in ToM (Schaafsma et al., 2015) or the specific ways
people infer others’ mental states (Quesque et al., 2024).”

3-2. Bayes Factor analysis:

In the explanation of the formula, I could not find an explanation of what the variable g reflects. It
appears the formula is integrating over a distribution of g . Is g a scaling parameter or distribution?

Thank you for pointing this out. In the Bayes Factor formula, g is a hyperparameter that
specifies prior variance of the effect size. Specifically, following Rouder et al. (2009), we placed
a prior of normal distribution on the standardized effect size. We assign its variance, g , an
inverse- $\chi^2(1)$ hyperprior, which can yield the recommended heavy-tailed (Cauchy) default on the
effect size when averaged across all possible values. When calculating the marginal likelihood
for the alternative hypothesis, we marginalize over the possible values of g to incorporate the
uncertainty about its scale. We have now clarified it on Page 37 as follows (highlighted in
green):

“In the formula, t represents the t -statistic, N the sample size, r the scale factor, and g the
prior variance of the standardized effect size... In estimating the likelihood for the
alternative hypothesis (the numerator), we marginalized over all possible values of g to
incorporate the uncertainty about the variance of the effect size (Rouder et al., 2009).”

Discussion:

3-3. Section 1:

The authors characterize the processes engaged by story reading/listening as "social interaction
perception". As they also discuss, the task-format used in the present study differs from other
studies presenting social interactions in the form of videos. I have the impression that the
interpretation of the study could benefit from a more elaborate definition of what the authors
understand as "social interaction". Did the authors also have researchers rate the videos for
"social interactions"? For example are acts of communication considered as a social interaction
in the same way as actions directed at another person (e.g. person A grabbing person B by the
arms, as in narrative #7 in the supplementary materials)? If yes, then it would be interesting to
know which of these types of social interaction are most common in the stories?

Thank you for the questions and the opportunity to provide further clarification. We defined
social interactions as story content in which more than one person is present, and one character
speaks to or directs actions toward other characters. We have now added this definition to Page
7 of the Main Text as follows (highlighted in green):

"We defined social interactions as parts of stories that have more than one character and in
which one character communicates with or directs actions toward other characters."

Several researchers in our group independently annotated social interactions in each
sentence part (as defined in Response 3-1; related results on Page 12 of the Main Text and Pages
1 and 6 of the Supplementary Materials). We ultimately decided that ratings from a group of
participants who were instructed to use the same criteria as the researchers were likely to be a
better normative measurement of the social interaction contents. Also, participants' continuous
ratings were highly reliable (mean split-half correlations = .98), comparable to those used for ToM,
and more suitable for comparing their neural correlates. Thus, in the Main Text we compared the
neural correlates of continuous social interaction ratings, rather than researcher annotations, with
continuous ToM ratings.

Based on this definition, our social interaction ratings and annotations involve both acts of
communication (or "character speaking") and actions directed at another person (or "physical
interactions"). We further manually annotated the two types of social interactions in each sentence
part to assess how common they are in our narratives. For sentence parts that involve both types
(e.g., "She ran to him and said that ..."), we annotated them as character speaking because the
acts and content of communications usually occupy most of the sentences, and this choice was
consistent with the annotations done in a previous study (Wolf et al., 2018). Overall, we found
that about 25.9% of all story time points contained character speaking, and another 22.7%
contained physical interactions. Thus, the two types of social interactions are about equally
common in our narratives. It is worth noting that they seemed to be processed in overlapping but

also separable brain regions engaged by social interaction processing. We have added this
information to Page 12 of the Main Text as follows (highlighted in green):

“Annotations also served to break down the social interaction content into approximately
equal amounts of verbal interactions and physical, non-verbal interactions (Wolf et al., 2018;
McMahon et al., 2023), suggesting relatively balanced interaction types (see Supplementary
Results).”

We have added the details mentioned above--how we annotated character speaking and
physical interactions, their correlations with social interactions and ToM, and their neural
correlates--to Pages 23-24 of the Supplementary Materials.

3-4. Section 3:

The authors note that several areas such as right aIPS, left PMC, bilateral LOTC or MT+ were
associated with tom ratings. These areas are linked to "mirror" processes and action
understanding.

I wonder if the authors can rule out alternative interpretations? For example, could low
level stimulus features explain differences in brain activation? Did the authors, for example, check
if sections of the narratives tagged as social-interaction-relevant by participants differ from
sections of the narratives tagged as tom-relevant in low level stimulus characteristics, such as
linguistic or semantic features? Linguistic features such as reading ease, word frequency, etc
could be checked, as well as more "natural language processing" type of analyses, such as
sentiment analyses, word semantic categorizations, or sentence complexity analyses. Many of
these types of analyses could be done without testing new participants, and if the authors could
show that there are no differences between story segments most highly rated for social-
interaction-relevance vs. tom-relevance, also no new analyses on the neural level would need to
be done. I think these kind of control analyses would significantly improve the interpretability of the
results, and in particular the interpretation of the additional network the authors find for tom
compared to social interactions.

Moreover, additional linguistic analyses or annotations could also allow the authors to
characterize the features associated with participants ratings of "social interaction". For example,
does that imply that action verbs were contained in the stories? Or that multiple persons were
featured in a sentence? Is that the same for tom-ratings? Just to give one example, a classic
study (Deen & McCarthy, 2010) found increased activation in the LOTC when participants read
stories featuring human movement, e.g. "running down the stairs", compared to other stories.

Deen, B., and McCarthy, G. (2010). Reading about the actions of others: biological motion
imagery and action congruency influence brain activity. *Neuropsychologia* 48, 1607–1615.

We appreciate the reviewer’s comments and the opportunity to validate and better
interpret our findings. For the lower-level linguistic features mentioned by the reviewer, we have
now annotated various stimulus features of our narratives using computational linguistic models,
including reading ease (word frequency and number of syllables), part-of-speech, sentence
complexity, and semantic frames. For higher-level features that can aid interpretations, we
considered different types of actions (intentional actions, all actions, and biological motion), the
presence of multiple people, indoor versus outdoor scenes, sentiment polarity (see Response 2-
1 for details), character speaking, and physical interactions (see Response 3-3). We compared
each annotated feature with the ratings of social interactions and ToM to test whether any features
could explain our findings, especially the voxels responding to ToM but not social interactions
(“ToM-only” regions, Figure 6). These new analyses also helped to better characterize the
features of social interaction and ToM that drive participants’ ratings.

In summary, we found that almost all lower-level features were uncorrelated with both
social interaction and ToM ratings (absolute $r_s < .1$, $r^2 < .01$). The presence of action verbs and
higher-level action-related features were also uncorrelated with social interaction and ToM ratings
(absolute $r_s \leq .14$, $r^2 \leq .02$). The only linguistic feature that had a nontrivial correlation with
social interactions/ToM was the semantic frame “Statement”, which is a specific type of social
interactions that largely overlaps with “character speaking” and conceptually related to social
interaction processing (addressed in detail in Response 3-3). These are summarized in new Table
S2 and Table S4 (reproduced below). Besides, none of the features tested reliably activated ToM-
selective voxels across modalities, and almost all of them had smaller effect sizes than ToM
(Figure S14 and Table S5, reproduced below; more details at the end of this response),
suggesting a specificity of those voxels responding only to ToM. One feature that correlated
strongly with social interactions, as expected, was the presence of multiple people in a scene
($r_s > .7$), and it correlated moderately with ToM (r_s approximately .3). However, we view the
presence of multiple people as a defining feature of social interactions rather than a confound. In
addition, it could not explain away the effects of social interactions or ToM on the brain (see
Response 2-1 for details). Together, these results indicate that the social-interaction-relevance
and ToM-relevance of the stories did not systematically differ in linguistic features or action-
related features, and these features are not alternative explanations of the neural correlates of
social interaction processing or ToM. In addition to these results, we will make the full set of
annotations available to the research community as a supplementary file upon publication, which

can be paired with the open fMRI dataset (Jung et al., 2025) and enable replications or further
 analyses.

We have added related contents to Pages 18-19 (highlighted in yellow) and 20 (highlighted
 in green) of the Main Text, and Pages 8-29 of the Supplementary Materials. Below, we provide
 details about how we defined and annotated each linguistic feature, their relationships with social
 interaction and ToM ratings, and how ToM-only regions responded to other features (all in the
 Supplementary Materials as well).

**Table S2**

*Pearson’s correlations between social cognitive features and ratings of social interactions*
 *and ToM across time*

Features	Social interactions			ToM		
	Audio	Text	All	Audio	Text	All
Intentional actions	-.02	.11	.06	.03	.03	.03
All actions	-.10	.14	.06	.02	-.03	-.01
Biological motion	-.05	.09	.04	.11	-.07	-.003
Multi-person presence	.70	.81	.77	.26	.29	.28
Indoor – outdoor	.03	.20	.14	-.05	.27	.16
“Positive” label	.01	-.08	-.04	-.01	.01	-.001
“Neutral” label	.17	-.09	-.004	.01	-.19	-.12
“Negative” label	-.23	.12	.03	-.002	.19	.13
“Positive” probabilities	.03	-.07	-.05	.03	-.03	.004
“Neutral” prob.	.10	-.08	.01	-.05	-.13	-.08
“Negative” prob.	-.14	.11	.02	.03	.14	.08

*Note.* “Text” and “Audio” indicate correlations within Text and Audio narratives, respectively, and
 “All” indicates correlations across all narratives.

**Table S4**

*All linguistic features annotated, the methods of annotations, and their Pearson’s*
 *correlations with ratings of social interactions and ToM*

Linguistic features	Annotation method (package used)	Social interactions			ToM		
		Audio	Text	All	Audio	Text	All
Reading ease – Word frequency	“wordfreq” in Python (Robyn Speer, 2022)	-.02	-.01	-.02	-.03	-.06	-.06

Reading ease – Number of syllables	“textstat” in Python ³	.03	.07	.06	.01	.04	.03
Part-of-speech (POS)	“flair” in Python (Akbik et al., 2018)	<.05	<.04	<.01	<.03	<.06	<.04
		.01 (all POS prediction)			.06 (all POS prediction)		
Sentence complexity	“textstat” in Python	.09	.03	.05	-.03	-.0003	.01
Semantic frames – “Statement”	“frame-semantic-transformer” in Python (Chanin, 2023)	.18	.39	.32	.16	.13	.14
Semantic frames – Other frames		<.12	<.09	<.09	<.11	<.13	<.12

Note. “Text” and “Audio” indicate correlations within Text and Audio narratives, respectively, and
“All” indicates correlations across all narratives. Cells with a “<” indicate that the correlations were
calculated between social interactions/ToM and more than one features (five POS tags and seven
frames) and the absolute values of all correlations were below the value shown there.

**Figure S14.** The effect sizes of ToM (light gray) and 10 other features in explaining neural activity
in the “ToM-only” voxels (Figure 6). The y axes showed the individual mean regression
coefficients (betas) divided by the standard deviation across participants. All other conventions
follow Figure S12. The “ToM-only” voxels were not reliably activated by other features across
modalities, and the effects of ToM were the strongest in both modalities.

**Table S5**

³ <https://pypi.org/project/textstat/>

Effect sizes of ToM and other social cognitive features in ToM-only voxels

	ToM	Intentional actions	All actions	Biological motion	Multi-person presence	Indoor versus outdoor	Positive versus Negative	Positive versus Neutral	Neutral versus Negative	Character speaking	Physical interaction
Text	0.76	0.12	0.03	-0.14	0.26	0.03	0.01	0.16	-0.21	-0.24	0.15
Audio	1.25	-0.19	-0.30	-0.11	0.50	0.02	-0.30	-0.34	-0.03	0.18	0.39
ToM-Text	0	0.64	0.73	0.62	0.50	0.73	0.75	0.60	0.55	0.53	0.61
ToM-Audio	0	1.07	0.95	1.14	0.75	1.23	0.96	0.91	1.22	1.07	0.86

*Note.* Bold fonts indicate values that are significantly deviating from zero under the Bonferroni
 Corrected (22 comparisons) threshold of .05 (p values estimated from 10,000 Bootstrap samples).
 In the first and second row, effect sizes of each feature in the Text and Audio modality were
 displayed. In the third and fourth row, the differences between the absolute effect sizes of ToM
 and those of each feature were in the Text and Audio modality were displayed; positive values
 indicate that ToM have larger absolute effect sizes.

Reading ease

We quantified the ease of reading narratives by two metrics: word frequency and number
 of syllables in each word. Word frequencies were found by the “wordfreq” python library (Robyn
 Speer, 2022), which returns a single frequency value for every English word based on various
 language sources. Word frequencies were unrelated to the medians of social interaction ratings
 ($r = -.02$) or ToM ratings ($r = -.06$) across all narratives used. Numbers of syllables of each word
 were counted by the “textstat” python library (<https://pypi.org/project/textstat/>) and were not
 related to social interaction ($r = .06$) or ToM ($r = .03$) ratings, either. To rule out the possibility that
 ratings of social interactions and ToM had a “lag” behind the words that triggered them due to
 participants’ response time, we incrementally shifted the time series of word frequencies and
 numbers of syllables backward and calculated the correlation for each shifted time series. We
 found that no shifted time series yielded larger correlations than the original ones.

Part-of-speech

We used the “flair” package in python (Akbik et al., 2018) to predict the part-of-speech
 (POS) label for each word in the narratives. Of the 30 unique POS tags predicted (for a full list
 see Table S3), most were relatively rare and could not support reliable estimations of their
 correlations with social interactions/ToM ratings. Thus, for individual POS, we only considered

three tags that were found in at least 10% of all words (“NN”, “IN”, and “VBD”) and two larger tag
groups: (1) “all action verbs”, defined as the union of the tags of “VB”, “VBD”, “VBG”, “VBN”, “VBP”,
“VBZ”, excluding the “be” verbs (“am”, “is”, “are”, “was”, “were”, “be”, “being”, and “been”); (2) “all
nouns” are the union of “NN”, “NNP”, “NNS”, and “NNPS”. Across all narratives, action verbs were
not correlated with either social interaction ($r = .01$) or ToM ($r = .03$) ratings ($r = .07$ at maximum
with time lags); the other four individual tags also correlated weakly with medians of social
interaction ratings (all absolute r s $< .01$) or ToM ratings (all absolute r s $< .04$). Besides individual
POS tags, we trained linear regression models that used all POS tags to predict social
interaction/ToM ratings in seven out of eight narratives, and tested model performance in the left-
out narrative. Across eight training-testing folds, the predicted ratings from POS tags correlated
weakly with the true social interaction ratings (average $r = .01$ (range [-.09, .07]) and ToM ratings
(average $r = .06$, range [-.11, .17]). Those results suggest that POS cannot reliably explain the
variance of social interaction or ToM ratings.

Active/Passive voice

We manually annotated whether each action in the narratives was written in the active or
passive voice. We found that a vast majority (96.1%) of actions in the narratives were in the active
voice, so it is not feasible to get reliable estimates of the relationships between active/passive
voice and social interaction/ToM ratings.

Sentence complexity

We used Flesch-Kincaid Grade Level (Kincaid et al., 1975) calculated by the “textstat”
library in python to quantify the complexity of each sentence in our narratives. Across all narratives,
the score did not correlate meaningfully with social interactions ($r = .05$) or ToM ($r = .01$) ratings.

Semantic frames

For semantic features in narratives, we evaluated the semantic “frames” defined by the
FrameNet project (Baker et al., 1998): A frame is a conceptual structure that describes a type of
situation, object, or event along with its participants and props. We used a transformer-based
model (Chanin, 2023) to automatically identify frames in our narratives. We found 275 unique
frames in all narratives we used, most of which appeared only once (42.5%) or twice (19.27%).
To find reliable relationships between any frames and social interaction/ToM ratings, we only
considered eight frames that appeared in at least 10% of sentences. We found that the frame
“Statement” was moderately correlated with social interaction ratings across all narratives ($r = .32$),

but relatively weakly with ToM ($r = .14$). All the absolute values of other correlations were below
0.15. The frame “Statement” was defined as containing verbs and nouns that communicated the
act of a Speaker to address a Message to some Addressee using language. This frame was
conceptually very similar to “Character speaking”, one specific type of social interactions
addressed in Response 3-3 (there is only one exception in our narratives, though, where a
character left a note to communicate some messages, which is using language without
interactions). Thus, the frame “Statement” should not be seen as a lower-level stimulus feature
or alternative explanation for the effects of social interactions.

Responses of ToM-only regions to other features

To test whether any features other than social interactions and ToM could explain neural
activity in ToM-only voxels, we calculated an effect size within those voxels for each feature. We
compared the absolute value of other effect sizes to zero and to those of ToM and estimated the
significance level by bootstrap tests, as we did for other effect size comparisons (e.g.,
comparisons between ToM demands and ToM engagement (Figure 4b)). Results showed that
almost all effect sizes tested were not significantly deviating from zero (all $ps > .05$ under
Bonferroni correction) except multi-person presence, positive versus negative sentences, and
physical interactions in the Audio modality. In both Text and Audio modality, ToM was the feature
associated with the largest effect size (Cohen’s $ds = .76, 1.25$, respectively), significantly larger
than the absolute values of all other effect sizes (all differences in Cohen’s $d > .50$, all corrected
$ps < .05$) except character speaking under Text modality (corrected $p = .09$). Those results support
that ToM-only regions were selectively activated by ToM but not other features tested.

References

- Akbik, A., Blythe, D. & Vollgraf, R. (2018). Contextual String Embeddings for Sequence
Labeling. *Proceedings of the 27th International Conference on Computational*
*Linguistics*, 1638–1649. <https://aclanthology.org/C18-1139/>
- Arioli, M. & Canessa, N. (2019). Neural processing of social interaction: Coordinate-based
meta-analytic evidence from human neuroimaging studies. *Human Brain Mapping*,
*40*(13), 3712–3737. <https://doi.org/10.1002/hbm.24627>
- Apperly, I. A. & Butterfill, S. A. (2009). Do humans have two systems to track beliefs and belief-
like states? *Psychological Review*, *116*(4), 953–970. <https://doi.org/10.1037/a0016923>

Baker, C. F., Fillmore, C. J. & Lowe, J. B. (1998). The Berkeley FrameNet Project. *Proceedings*
*of the 17th International Conference on Computational Linguistics* -. the 17th
international conference, Montreal, Quebec, Canada.
<https://doi.org/10.3115/980451.980860>

Bannert, M. M. & Bartels, A. (2018). Human V4 activity patterns predict behavioral performance
in imagery of object color. *The Journal of Neuroscience: The Official Journal of the*
*Society for Neuroscience*, 38(15), 3657–3668.
<https://doi.org/10.1523/JNEUROSCI.2307-17.2018>

Cinelli, C., Forney, A. & Pearl, J. (2024). A crash course in good and bad controls. *Sociological*
*Methods & Research*, 53(3), 1071–1104. <https://doi.org/10.1177/00491241221099552>

Clark, A. (2013). Whatever next? Predictive brains, situated agents, and the future of cognitive
science. *The Behavioral and Brain Sciences*, 36(3), 181–204.
<https://doi.org/10.1017/S0140525X12000477>

Chanin, D. (2023). Open-source frame semantic parsing. In *arXiv [cs.CL]*. arXiv.
<http://arxiv.org/abs/2303.12788>

Chen, J., Leong, Y. C., Honey, C. J., Yong, C. H., Norman, K. A. & Hasson, U. (2017). Shared
memories reveal shared structure in neural activity across individuals. *Nature*
*Neuroscience*, 20(1), 115–125. <https://doi.org/10.1038/nn.4450>

Deen, B. & McCarthy, G. (2010). Reading about the actions of others: biological motion imagery
and action congruency influence brain activity. *Neuropsychologia*, 48(6), 1607–1615.
<https://doi.org/10.1016/j.neuropsychologia.2010.01.028>

Desmet, C. & Brass, M. (2015). Observing accidental and intentional unusual actions is
associated with different subregions of the medial frontal cortex. *NeuroImage*, 122, 195–
202. <https://doi.org/10.1016/j.neuroimage.2015.08.018>

Iacoboni, M., Molnar-Szakacs, I., Gallese, V., Buccino, G., Mazziotta, J. C. & Rizzolatti, G. (2005).
Grasping the intentions of others with one's own mirror neuron system. *PLoS Biology*, 3(3),
e79. <https://doi.org/10.1371/journal.pbio.0030079>

Heider, F. & Simmel, M. (1944). An Experimental Study of Apparent Behavior. *The American*
*Journal of Psychology*, 57(2), 243–259. <https://doi.org/10.2307/1416950>

Isik, L., Koldewyn, K., Beeler, D. & Kanwisher, N. (2017). Perceiving social interactions in the
posterior superior temporal sulcus. *Proceedings of the National Academy of Sciences of*
*the United States of America*, 114(43), E9145–E9152.
<https://doi.org/10.1073/pnas.1714471114>

- Jacoby, N., Bruneau, E., Koster-Hale, J. & Saxe, R. (2016). Localizing Pain Matrix and Theory
of Mind networks with both verbal and non-verbal stimuli. *NeuroImage*, 126, 39–48.
<https://doi.org/10.1016/j.neuroimage.2015.11.025>
- Jung, H., Amini, M., Hunt, B. J., Murphy, E. I., Sadil, P., Halchenko, Y. O., Petre, B., Miao, Z.,
Kragel, P. A., Han, X., Heilicher, M. O., Sun, M., Collins, O. G., Lindquist, M. A. & Wager,
972 T. D. (2025). Spacetop: A multimodal fMRI dataset unifying naturalistic processes with a
973 rich array of experimental tasks. *Scientific Data*, 12(1), 1–18.
<https://doi.org/10.1038/s41597-025-05154-x>
- Kana, R. K., Ammons, C. J., Doss, C. F., Waite, M. E., Kana, B., Herringshaw, A. J. & Ver Hoef,
976 L. (2015). Language and motor cortex response to comprehending accidental and
977 intentional action sentences. *Neuropsychologia*, 77, 158–164.
<https://doi.org/10.1016/j.neuropsychologia.2015.08.020>
- Kincaid, J. P., Fishburne, R. P., Rogers, R. L., & Chissom, B. S. (1975). *Derivation of new*
*readability formulas (automated readability index, fog count, and flesch reading ease*
*formula) for Navy enlisted personnel* (Report No. RBR-8-75). Naval Air Station Memphis,
TN: Chief of Naval Technical Training, Research Branch.
- Kragel, P. A., Reddan, M. C., LaBar, K. S. & Wager, T. D. (2019). Emotion schemas are
embedded in the human visual system. *Science Advances*.
<https://www.science.org/doi/epdf/10.1126/sciadv.aaw4358>
- Lahnakoski, J. M., Glerean, E., Salmi, J., Jääskeläinen, I. P., Sams, M., Hari, R. & Nummenmaa,
987 L. (2012). Naturalistic fMRI mapping reveals superior temporal sulcus as the hub for the
988 distributed brain network for social perception. *Frontiers in Human Neuroscience*, 6, 233.
<https://doi.org/10.3389/fnhum.2012.00233>
- Landsiedel, J., Daughters, K., Downing, P. E. & Koldewyn, K. (2022). The role of motion in the
neural representation of social interactions in the posterior temporal cortex. *NeuroImage*,
262, 119533. <https://doi.org/10.1016/j.neuroimage.2022.119533>
- Landsiedel, J. & Koldewyn, K. (2023). Auditory dyadic interactions through the “eye” of the
social brain: How visual is the posterior STS interaction region? *Imaging Neuroscience*
*(Cambridge, Mass.)*, 1, 1–20. https://doi.org/10.1162/imag_a_00003
- Lee Masson, H. & Isik, L. (2021). Functional selectivity for social interaction perception in the
human superior temporal sulcus during natural viewing. *NeuroImage*, 245, 118741.
<https://doi.org/10.1016/j.neuroimage.2021.118741>
- Lee Masson, H., Chen, J. & Isik, L. (2024). A shared neural code for perceiving and
remembering social interactions in the human superior temporal sulcus.

*Neuropsychologia*, 196(108823), 108823.
<https://doi.org/10.1016/j.neuropsychologia.2024.108823>

Lee, S.-H., Kravitz, D. J. & Baker, C. I. (2012). Disentangling visual imagery and perception of
real-world objects. *NeuroImage*, 59(4), 4064–4073.
<https://doi.org/10.1016/j.neuroimage.2011.10.055>

Liang, M., Mouraux, A., Hu, L. & Iannetti, G. D. (2013). Primary sensory cortices contain
distinguishable spatial patterns of activity for each sense. *Nature Communications*, 4(1),
1979. <https://doi.org/10.1038/ncomms2979>

Mangnus, M., Koch, S. B. J., Cai, K., Romaneli, M. G., Hagoort, P., Bašnáková, J. & Stolk, A.
(2024). Preserved spontaneous mentalizing amid reduced intersubject variability in
autism during a movie narrative. *Biological Psychiatry: Cognitive Neuroscience and*
*Neuroimaging*. <https://doi.org/10.1016/j.bpsc.2024.10.007>

McMahan, E., Bonner, M. F. & Isik, L. (2023). Hierarchical organization of social action features
along the lateral visual pathway. *Current Biology: CB*, 33(23), 5035-5047.e8.
<https://doi.org/10.1016/j.cub.2023.10.015>

McMahan, E. & Isik, L. (2023). Seeing social interactions. *Trends in Cognitive Sciences*.
<https://doi.org/10.1016/j.tics.2023.09.001>

Mele, A. R. & Cushman, F. (2007). Intentional action, folk judgments, and stories: Sorting things
out. *Midwest Studies in Philosophy*, 31(1), 184–201. [https://doi.org/10.1111/j.1475-](https://doi.org/10.1111/j.1475-4975.2007.00147.x)
[4975.2007.00147.x](https://doi.org/10.1111/j.1475-4975.2007.00147.x)

Moro, V., Berlucchi, G., Lerch, J., Tomaiuolo, F. & Aglioti, S. M. (2008). Selective deficit of
mental visual imagery with intact primary visual cortex and visual perception. *Cortex; a*
*Journal Devoted to the Study of the Nervous System and Behavior*, 44(2), 109–118.
<https://doi.org/10.1016/j.cortex.2006.06.004>

Nijhof, A. D. & Willems, R. M. (2015). Simulating fiction: individual differences in literature
comprehension revealed with fMRI. *PloS One*, 10(2), e0116492.
<https://doi.org/10.1371/journal.pone.0116492>

Pearl, J. (2021). Causal and Counterfactual Inference. In M. Knauff & W. Spohn (Eds.), *The*
*Handbook of Rationality* (pp. 427–438). The MIT Press.

Pelphrey, K. A., Morris, J. P. & McCarthy, G. (2004). Grasping the intentions of others: the
perceived intentionality of an action influences activity in the superior temporal sulcus
during social perception. *Journal of Cognitive Neuroscience*, 16(10), 1706–1716.
<https://doi.org/10.1162/0898929042947900>

- Pérez, J. M., Rajngewerc, M., Giudici, J. C., Furman, D. A., Luque, F., Alemany, L. A. & Martínez,
1035 M. V. (2021). pysentimiento: A Python Toolkit for Opinion Mining and Social NLP tasks. In
*arXiv [cs.CL]*. arXiv. <http://arxiv.org/abs/2106.09462>
- Quesque, F., Apperly, I., Baillargeon, R., Baron-Cohen, S., Becchio, C., Bekkering, H.,
Bernstein, D., Bertoux, M., Bird, G., Bukowski, H., Burgmer, P., Carruthers, P., Catmur,
C., Dziobek, I., Epley, N., Erle, T. M., Frith, C., Frith, U., Galang, C. M., ... Brass, M.
(2024). Defining key concepts for mental state attribution. *Communications Psychology*,
2(1), 29. <https://doi.org/10.1038/s44271-024-00077-6>
- Rakoczy, H. (2022). Foundations of theory of mind and its development in early childhood.
*Nature Reviews Psychology*, 1(4), 223–235. <https://doi.org/10.1038/s44159-022-00037-z>
- Rao, R. P. & Ballard, D. H. (1999). Predictive coding in the visual cortex: a functional
interpretation of some extra-classical receptive-field effects. *Nature Neuroscience*, 2(1),
79–87. <https://doi.org/10.1038/4580>
- Richardson, H., Lisandrelli, G., Riobueno-Naylor, A. & Saxe, R. (2018). Development of the
social brain from age three to twelve years. *Nature Communications*, 9(1), 1027.
<https://doi.org/10.1038/s41467-018-03399-2>
- Robyn Speer. (2022). rspeer/wordfreq: v3.0 (v3.0.2). *Zenodo*.
<https://doi.org/10.5281/zenodo.7199437>
- Rouder, J. N., Speckman, P. L., Sun, D., Morey, R. D. & Iverson, G. (2009). Bayesian t tests for
accepting and rejecting the null hypothesis. *Psychonomic Bulletin & Review*, 16(2), 225–
237. <https://doi.org/10.3758/PBR.16.2.225>
- Schaafsma, S. M., Pfaff, D. W., Spunt, R. P. & Adolphs, R. (2015). Deconstructing and
reconstructing theory of mind. *Trends in Cognitive Sciences*, 19(2), 65–72.
<https://doi.org/10.1016/j.tics.2014.11.007>
- Schneider, D., Slaughter, V. P. & Dux, P. E. (2017). Current evidence for automatic Theory of
Mind processing in adults. *Cognition*, 162, 27–31.
<https://doi.org/10.1016/j.cognition.2017.01.018>
- Sliwa, J. & Freiwald, W. A. (2017). A dedicated network for social interaction processing in the
primate brain. *Science*, 356(6339), 745–749. <https://doi.org/10.1126/science.aam6383>
- Varrier, R. S. & Finn, E. S. (2022). Seeing social: A neural signature for conscious perception of
social interactions. *The Journal of Neuroscience: The Official Journal of the Society for*
*Neuroscience*. <https://doi.org/10.1523/JNEUROSCI.0859-22.2022>
- Wagner, D. D., Kelley, W. M., Haxby, J. V. & Heatherton, T. F. (2016). The Dorsal Medial
Prefrontal Cortex Responds Preferentially to Social Interactions during Natural Viewing.

*The Journal of Neuroscience: The Official Journal of the Society for Neuroscience*,
36(26), 6917–6925. <https://doi.org/10.1523/JNEUROSCI.4220-15.2016>

Wolf, D., Mittelberg, I., Reikittke, L.-M., Bhavsar, S., Zvyagintsev, M., Haeck, A., Cong, F.,
Klasen, M. & Mathiak, K. (2018). Interpretation of Social Interactions: Functional Imaging
of Cognitive-Semiotic Categories During Naturalistic Viewing. *Frontiers in Human*
*Neuroscience*, 12, 296. <https://doi.org/10.3389/fnhum.2018.00296>

Yeo, B. T. T., Krienen, F. M., Sepulcre, J., Sabuncu, M. R., Lashkari, D., Hollinshead, M.,
Roffman, J. L., Smoller, J. W., Zöllei, L., Polimeni, J. R., Fischl, B., Liu, H. & Buckner, R.
1076 L. (2011). The organization of the human cerebral cortex estimated by intrinsic functional
connectivity. *Journal of Neurophysiology*, 106(3), 1125–1165.
<https://doi.org/10.1152/jn.00338.2011>

Reviewer #1 (Remarks to the Author):

The authors have addressed the points I raised thoroughly and satisfactorily. They have also worked hard to address the points raised by the other reviewers. As far as I am concerned I consider this revised version acceptable for publication.

I have one suggestion. It would be very helpful to me and, I expect, to other readers if the authors could give examples in the main text of sentences which were: 1) social interaction only, 2) ToM only, and 3) both.

We are grateful for the reviewer's feedback. We have now incorporated examples of narrative sentences that have high ratings for only social interactions, only theory of mind, and both, in the main text (highlighted on Page 14).

Reviewer #2 (Remarks to the Author):

The authors have addressed my concerns.

Reviewer #3 (Remarks to the Author):

All of my comments and questions have been appropriately addressed in this revised version of the manuscript. In particular, the following points have greatly improved the manuscript:

First, linguistic control analyses have been carried out for the stimulus material, and the results from these analyses show that alternative linguistic interpretations of the results (which I pointed out as a potential concern) are highly unlikely.

Second, the authors carried out additional annotations of the stimuli (in response to my and Reviewer #2's concerns), which further improve the interpretability and specificity of their results. The neural correlates of these additional annotations reveal the nuance

and complexity of the social stimulus material presented in this study, providing readers a deeper look into the neural representation of rich social interactions, going beyond label-based categorizations in terms of "theory of mind" and "social interactions".

Altogether, the authors made great efforts in revising and improving the manuscript, and in my judgment the manuscript is now in an acceptable form.